# Generative modeling for RNA splicing prediction and design

Di Wu[1], Natalie Maus[1], Anupama Jha[2], Kevin Yang[3], Benjamin D Wales-McGrath[3], San Jewell[3], Anna Tangiyan[4], Peter Choi[4,5], Jake R Gardner[1], Yoseph Barash[1,3]*

[1]Department of Computer and Information Science, School of Engineering, University of Pennsylvania, Philadelphia, United States; [2]Department of Genome Sciences, University of Washington, Seattle, United States; [3]Department of Genetics, Perelman School of Medicine, University of Pennsylvania, Philadelphia, United States; [4]Division of Cancer Pathobiology, The Children's Hospital of Philadelphia, Philadelphia, United States; [5]Department of Pathology & Laboratory Medicine, Perelman School of Medicine, University of Pennsylvania, Philadelphia, United States

## eLife Assessment

TrASPr is an **important** contribution that leverages transformer models focused on regulatory regions to enhance predictions of tissue-specific splicing events. The revisions strengthen the manuscript by clarifying methodology and expanding analyses across exon and intron sizes, and the evidence supporting TrASPr's predictive performance is **compelling**. This work will be of interest to researchers in computational genomics and RNA biology, offering an improved model for splicing prediction and a promising approach to RNA sequence design.

*For correspondence:
yosephb@biociphers.org

Competing interest: The authors declare that no competing interests exist.

**Abstract** Alternative splicing (AS) of pre-mRNA plays a crucial role in tissue-specific gene regulation, with disease implications due to splicing defects. Predicting and manipulating AS can therefore uncover new regulatory mechanisms and aid in therapeutic design. We introduce TrASPr+BOS, a generative AI model with Bayesian Optimization for predicting and designing RNA for tissue-specific splicing outcomes. Transformer for Alternative Splicing Prediction (TrASPr) is a multi-transformer model that can handle different types of AS events and generalize to unseen cellular conditions. It then serves as an oracle, generating labeled data to train a Bayesian Optimization for Splicing (BOS) algorithm to design RNA for condition-specific splicing outcomes. We show TrASPr+BOS outperforms existing methods, enhancing tissue-specific AUPRC by up to 1.8-fold and capturing tissue-specific regulatory elements. We validate hundreds of predicted novel tissue-specific splicing variations and confirm new regulatory elements using dCas13. We envision TrASPr+BOS as a light yet accurate method researchers can probe or adopt for specific tasks.

## Introduction

Alternative splicing (AS) occurs when different mature mRNA transcripts are produced from the same gene by selectively including or excluding specific pre-mRNA exonic or intronic segments (see *Figure 1a* for illustrative examples). Over 90% of human genes undergo AS, with conservative estimates suggesting that at least 35% of human genes switch their dominant isoform across 16 adult tissues (*Pan et al., 2008*; *Wang et al., 2008*; *Gonzàlez-Porta et al., 2013*). At the molecular level, AS can alter protein function, for instance, by removing a nuclear localization signal or modifying a binding domain within the encoded protein (*Smith and Valcárcel, 2000*; *Licatalosi and Darnell, 2010*). AS can also serve as a mechanism to control gene expression by introducing a pre-termination

codon (*Lewis et al., 2003*), and aberrant splicing has been implicated in numerous diseases (*Singh and Cooper, 2012*). Consequently, a long-term objective for the RNA community has been to develop a predictive 'splicing code' model capable of determining splicing outcomes based on genomic sequence and cell or tissue type (*Wang and Burge, 2008*). This long-term objective forms the focus of this work.

Decades of extensive research into the mechanisms of splicing have identified hundreds of RNA-binding proteins (RBPs) involved in splicing regulation. These RBPs bind to exons and nearby intronic regions, typically within a few hundred bases of proximal splice sites, to modulate splicing in a condition-specific manner (*Fu and Ares, 2014*). However, translating this 'parts list' of RBPs into a predictive splicing code (*Wang and Burge, 2008*) remains a significant challenge. One major techno-logical advancement that facilitated predictive splicing code developments is high-throughput RNA sequencing (RNA-Seq). RNA-Seq enabled researchers to detect and quantify thousands of AS events across diverse cellular conditions and tissues, as illustrated in *Figure 1b*. This advancement provided the data necessary to train predictive splicing codes using machine learning. Specifically, sequencing reads that span RNA segments joined by splicing (splice junction reads) are analyzed by dedicated algorithms such as MISO (*Katz et al., 2010*), MAJIQ (*Vaquero-Garcia et al., 2023*; *Vaquero-Garcia et al., 2016*), and rMATS (*Shen et al., 2014*) to quantify these AS events.

The quantification of AS events, expressed as 'percent spliced in' (PSI), represents the ratio of isoforms (supported by junction-spanning reads) that include or exclude a specific RNA segment. Formally, PSI for a cassette exon $e$ in cell type $c$ can be denoted $\Psi_{e,c} \in [0,1]$, and changes in splicing for that exon between two cell types $c, c'$ are denoted as dPSI $\Delta\Psi_{e,c,c'} \in [-1,1], \Delta\Psi_{e,c,c'} = \Psi_{e,c} - \Psi_{e,c'}$. Depending on the task, these $\Psi, \Delta\Psi$ values across numerous AS events and conditions can serve as labels to train machine learning algorithms.

As high-throughput splicing measurements became widely accessible, researchers were able to define a variety of prediction tasks based on this data. For instance, the first splicing code focused on qualitative changes in inclusion levels ('up', 'down', or 'no change') for cassette exons across tissues, identifying associated regulatory mechanisms (*Barash et al., 2010*). Subsequent studies shifted to related tasks, such as predicting the effects of genetic mutations on cassette exon inclusion (*Xiong et al., 2015*; *Cheng et al., 2021*) or the creation and disruption of splice sites caused by mutations (*Jaganathan et al., 2019*; *Zeng and Li, 2022*). In this work, we address two tasks: quantitative tissue-specific splicing prediction for a given AS event and splicing sequence design. We demonstrate how these tasks are interconnected through deep generative models and illustrate their utility. We now proceed to describe each task and the related work in detail.

The tissue-specific prediction task involves predicting $\Psi_{e,c}$, $\Delta\Psi_{e,c,c'}$ given the tissue or cell type and the genomic sequence of an AS event (e.g. cassette exon $e$). Early splicing code models relied on manually curated regulatory features from the literature to predict tissue-specific splicing changes (*Barash et al., 2010*; *Xiong et al., 2015*; *Xiong et al., 2011*). With advances in high-throughput splicing quantification, these earlier models—based on boosted decision trees and Bayesian neural networks—were replaced by deep neural networks, such as AutoEncoders, long short-term memory (LSTM) networks, and convolutional neural networks (CNNs; *Jha et al., 2017*; *Bretschneider et al., 2018*; *Cheng et al., 2021*; *Zeng and Li, 2022*).

Recent years saw a shift from using predefined features and relatively small models to large models that scan wide windows of genomic sequences, typically for predicting the effect of genetic variants on splicing outcomes. Such models do not assume a specific AS event type (e.g. cassette exon) but instead look for changes in splicing within a genomic window. Most notable is SpliceAI (*Jaganathan et al., 2019*), which uses a CNN ResNet architecture, scanning 10 kilobases (kb) size windows across the genome to predict whether the center position is a splice site. More recently, Pangolin used the SpliceAI model architecture but trained it on several tissues and four species to predict tissue-specific PSI values ($\Psi_{e,c}$) (*Zeng and Li, 2022*). The emphasis on large models scanning genomic windows has extended to recent Large Language Model (LLM)-based approaches for various genomic tasks. Examples include DNABERT (*Ji et al., 2021*), which pre-trains a BERT model on human DNA, SPLICE-BERT (*Chen et al., 2023*), which incorporates evolutionary conservation for splice-site prediction, and Enformer (*Avsec et al., 2021*), which predicts gene expression across 200 kb windows. A notable recent addition is SpliceTransformer (*You et al., 2024*), which we compare against and discuss in greater detail in the discussion section.

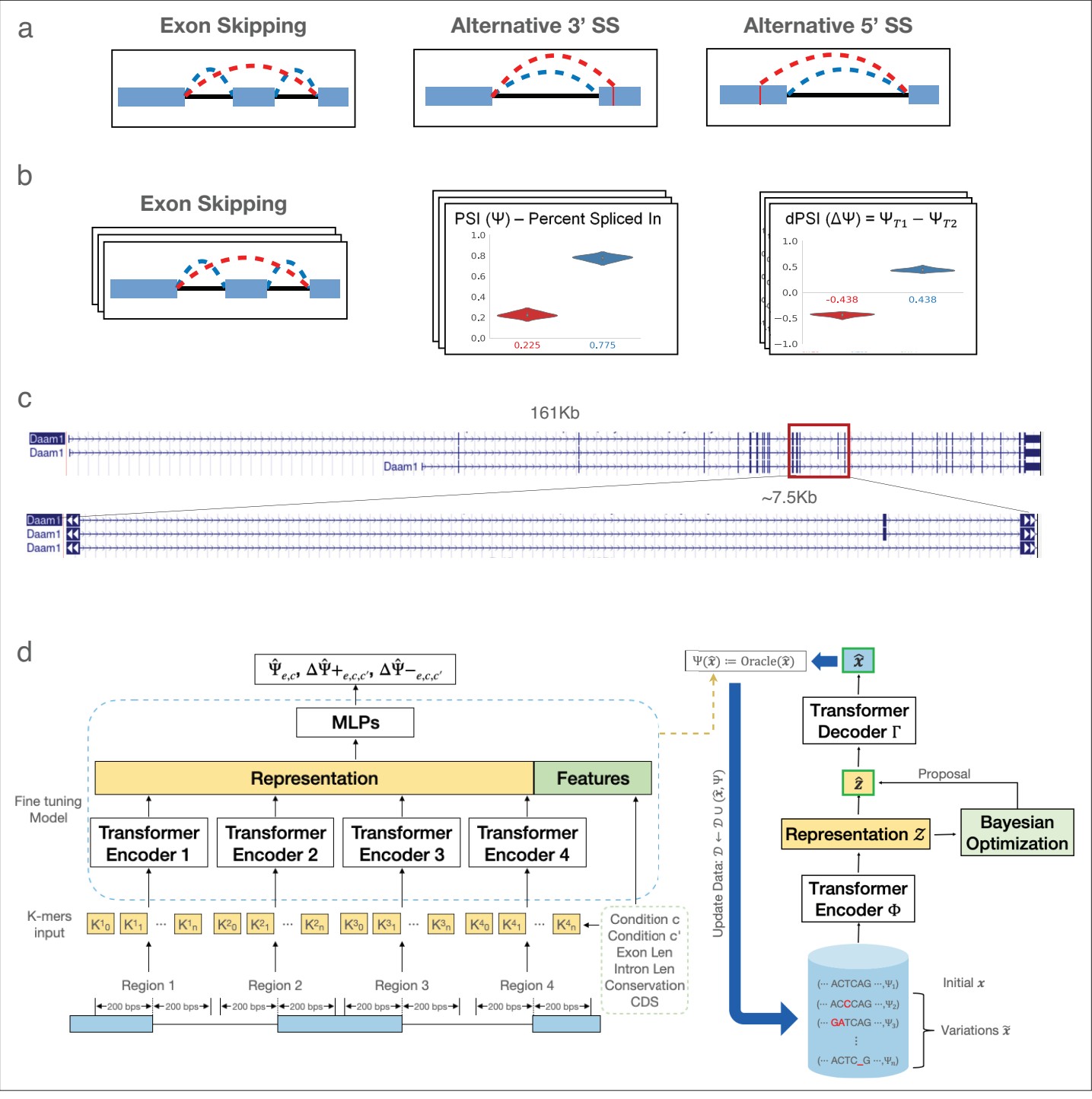

**Figure 1.** RNA alternative splicing (AS) and its predictive generative modeling. (**a**) Cartoons of several types of AS, involving exon skipping as well as alternative 3' and 5' splice sites. (**b**) Quantification of AS events such as exon skipping from RNA-Seq. PSI is used to represent the relative exon inclusion level, and dPSI is used to show the inclusion change across different conditions. Quantifying thousands of such AS events across many conditions, represented by the 3D stacks, enables training ML algorithms to predict PSI and dPSI from genomic sequence. (**c**) A genome browser view of an illustrative exon skipping event. The genomic regions spanned by cassette exons vary from tens to hundreds of thousands of bases. (**d**) Schematic of TrASPr (left) and BOS (right). TrASPr's input includes genomic regions proximal to splice sites around AS event $e$, along with other genomic features and two cellular conditions ($c, c'$; bottom left). It outputs a predicted inclusion level $\hat{\Psi}_{e,c}$ as well as predictions for increased inclusion and increased exclusion between the two cellular conditions ($\Delta\hat{\Psi}+_{e,c,c'}, \Delta\hat{\Psi}-_{e,c,c'}$). Its predictions are then used to train the generative model using BOS. See main text for details.

The tissue-specific splicing prediction task described above, as well as the application of LLM models to it, presents several challenges. For example, the Transformer model is a neural architecture based on self-attention that excels at modeling long-range dependencies in sequence data (*Vaswani et al., 2017*), which has shown strong performance in various genomic tasks. However, genomic regions containing cassette exons vary greatly in size, from a few hundred bases to several kilobases. Thus, adopting 'out-of-the-box' transformer-based models like DNABERT (*Ji et al., 2021*), which are limited to capturing sequences up to ~500 bp in length, is unsuitable for cassette exons. Even larger models like SpliceAI and Pangolin, which are highly effective at splice site detection, span no more than 10 kb. In contrast, 24% of the cassette exons analyzed in this study span a region between the flanking exons' upstream 3' and downstream 5' splice sites that are larger than 10 kb. Additionally, such regions may involve multiple exons and junctions, complicating the condition-specific prediction of PSI or dPSI. Another significant challenge is sample size. For any pair of cell types or tissues $(c, c')$, the number of cassette exons exhibiting significant inclusion changes is typically limited to a few hundred. Moreover, the RBPs responsible for these changes are drawn from a pool of hundreds encoded in the human genome, and their specific contributions vary. Further complicating the task is the poorly defined nature of RBP binding sites, which have low information content and can either enhance or repress exon inclusion depending on their binding position (*Fu and Ares, 2014*).

The second task we address in this work is splicing sequence design. This task can be defined as generating a genomic sequence $S$ with a specific splicing profile across various conditions $\{c\}$. For example, the goal may be to create a cassette exon $e$ that is highly included in the cerebellum but minimally included in other tissues. Importantly, the genomic sequence $S$ may include intronic regions containing regulatory elements that influence the splicing of exon $e$. The sequence can also be based on an existing exon $e$ that requires enhancement or correction through genetic editing. The allowed editing 'budget'—such as the number of edits or total base changes—can be specified by the user to meet the requirements of a particular task. This flexibility is useful for applications like CRISPR editing or modifying exon inclusion using antisense oligonucleotides (ASOs) for therapeutic purposes. For instance, the model could be tasked with designing a new version of the cassette exon, restricted to no more than $N$ edit locations (i.e. start position of one or more consecutive bases) and $M$ total base changes.

The splicing sequence design task is much newer than the tissue-specific splicing prediction task, with few related works published so far. Specifically, for RNA design, prior work has focused on areas such as optimizing untranslated regions (UTRs) in mRNA vaccines for enhanced expression (*Castillo-Hair and Seelig, 2022*; *Leppek et al., 2022*) and designing alternative polyadenylation sequences (*Bogard et al., 2019*). The algorithms employed in these studies include genetic algorithms for 5' UTR design (*Sample et al., 2019*) and Deep Exploration Networks (DEN; *Linder et al., 2019*). Researchers also used a greedy evolution approach where they introduced mutations one at a time to a given sequence and assessed the outcome using SpliceAI (*Wilkins et al., 2024*). However, none of these works defined the tissue-specific splicing design task for cassette exons as formulated here.

In the first part of this work, we address the challenges of tissue-specific splicing predictions by developing TrASPr (Transformer for Alternative Splicing Prediction), a multi-transformer-based model illustrated in *Figure 1d* (left). Compared to existing models, TrASPr takes a distinct approach. While TrASPr uses the Transformer model architecture common in LLMs, it does not involve billions of parameters. It also does not scan long genomic windows as common in methods such as SpliceAI and many LLM 'foundation' models nowadays. Instead, TrASPr uses approximately 189 M parameters and leverages existing transcriptome annotations along with current knowledge of splicing regulation to focus it on specific genomic regions. Decades of research indicate that splicing regulation is primarily localized around splice sites and influenced by condition-specific RBPs that mediate competition between splice sites. Based on this insight, we first pre-train a transformer to recognize splice sites. We then center a dedicated transformer around each of the splice sites of the cassette exon and its upstream and downstream (competing) exons (four separate transformers for four splice sites in total).

To account for features not captured by the local genomic sequence (e.g. intron and exon length, conservation), we incorporate additional genomic features. The representations learned by each transformer are combined using several joint multilayer perceptron (MLP) layers. This integrated model, TrASPr, is then trained on specific AS events (e.g. cassette exons) detected and quantified from the human transcriptome across different tissues.

By leveraging transcriptome annotations and quantifications, TrASPr can handle cassette exons spanning a wide range of window sizes—from 181 to 329,227 bases—thanks to its multi-transformer architecture. We first evaluate TrASPr using RNA splicing data from six human tissues (genotype-tissue expression [GTEx] dataset), where it achieves state-of-the-art (SOTA) prediction accuracy compared to existing models. We also perform ablation studies to assess alternative architectures.

Next, we demonstrate TrASPr's ability to detect regulatory elements, including tissue-specific ones, using diverse datasets. These include datasets involving four RBP knockdowns (KDs), a high-throughput mutagenesis assay introducing thousands of genetic variants, and a lower-throughput dataset for tissue-specific regulatory elements from a mini-gene reporter assay.

To illustrate the versatility of TrASPr, we showcase its application in two key tasks. First, we use it to predict novel tissue-specific splicing variations, which we confirm through targeted sequencing. Second, we validate TrASPr's predictions of specific regulatory elements by targeting them with dCas13d.

TrASPr's ability to predict tissue-specific splicing enables its use as a 'teacher' or an 'oracle' for training a second deep generative model, which addresses the novel task of RNA splicing sequence design. To this end, we employ a variational autoencoder (VAE) transformer generative model, training it to structure its latent space representation using Bayesian Optimization (BO) techniques. Our Bayesian Optimization for Splicing (BOS) algorithm optimizes the VAE transformer under user-defined sequence and splicing outcome constraints. We evaluate BOS-generated sequences against two baselines: random mutations and a previously proposed genetic algorithm. The results demonstrate that BOS more effectively mutates a given sequence, even with a limited number of mutations, to achieve a predefined tissue-specific splicing outcome. Additionally, we show that the mutations selected by BOS align well with experimentally validated mutations, providing further evidence of its utility. Finally, we conclude with a discussion of the potential applications of this approach, its limitations, and directions for future research.

## Results

### TrASPr offers improved prediction of cassette exon inclusion across diverse human tissues

To achieve accurate predictions for $\Psi, \Delta\Psi$ across diverse human tissues, we start by pre-training a 6-layer 12-head BERT model on annotated 3' and 5' human splice sites. We then employ four such BERT models in the architecture of TrASPr shown in *Figure 1d*, training it on cassette exons quantified across six human tissues using GTEx V8 (see Methods for details). To assess TrASPr prediction accuracy, we compared it to SpliceAI (*Jaganathan et al., 2019*), Pangolin (*Zeng and Li, 2022*), and the recent SpliceTransformer (*You et al., 2024*). SpliceAI is considered the current SOTA for predicting splice site strength, given a genomic window of 10 kb around it. While the model is agnostic to cassette exon definition or the cell type, it has been extensively used to approximate exon inclusion and predict mutations that disrupt splicing in patients. Pangolin adopted the SpliceAI model for tissue-specific splicing, training it on data from four species, each with four tissues (*Zeng and Li, 2022*). Specifically, Pangolin was trained on quantification of 3' and 5' splice site 'usage' as defined by SpliSER (*Dent et al., 2021*), which, for cassette exons, are equivalent to MAJIQ's LSV (local splicing variation) based PSI values. Finally, the most recent SpliceTransformer uses a modeling approach similar to Pangolin and SpliceAI, that is a larger genomic window (8000 bp) around a 3' or 5' splice site, but replaces the ResNet CNN with a longhorn Transformer, a unique tissue-specific splice site 'usage' defined by the authors, and a loss function that promotes splice sites with a tissue-specific 'usage'. To match cassette exon inclusion to SpliceAI, Pangolin, and SpliceTransformer's splice site scores, we found that averaging the scores for the alternative exon's 3' and 5' splice sites performed best and applied it to the evaluations described below (see Methods).

For all models, performance was evaluated on shared tissues and test chromosomes used by Pangolin (*Zeng and Li, 2022*; see Data for details). We assessed the performance of PSI prediction using two statistics: Pearson correlation ($r$) and the fraction of samples (denoted $a$) for which the model's PSI prediction was close to the observed value (<0.2 difference, i.e. within the white dashed lines in *Figure 2a*). Treating SpliceAI as a baseline for SOTA non-tissue-specific splice site strength prediction, we see in *Figure 2a* left column that it is able to achieve $a = 0.66, r = 0.67$. Its tissue-specific adaptation

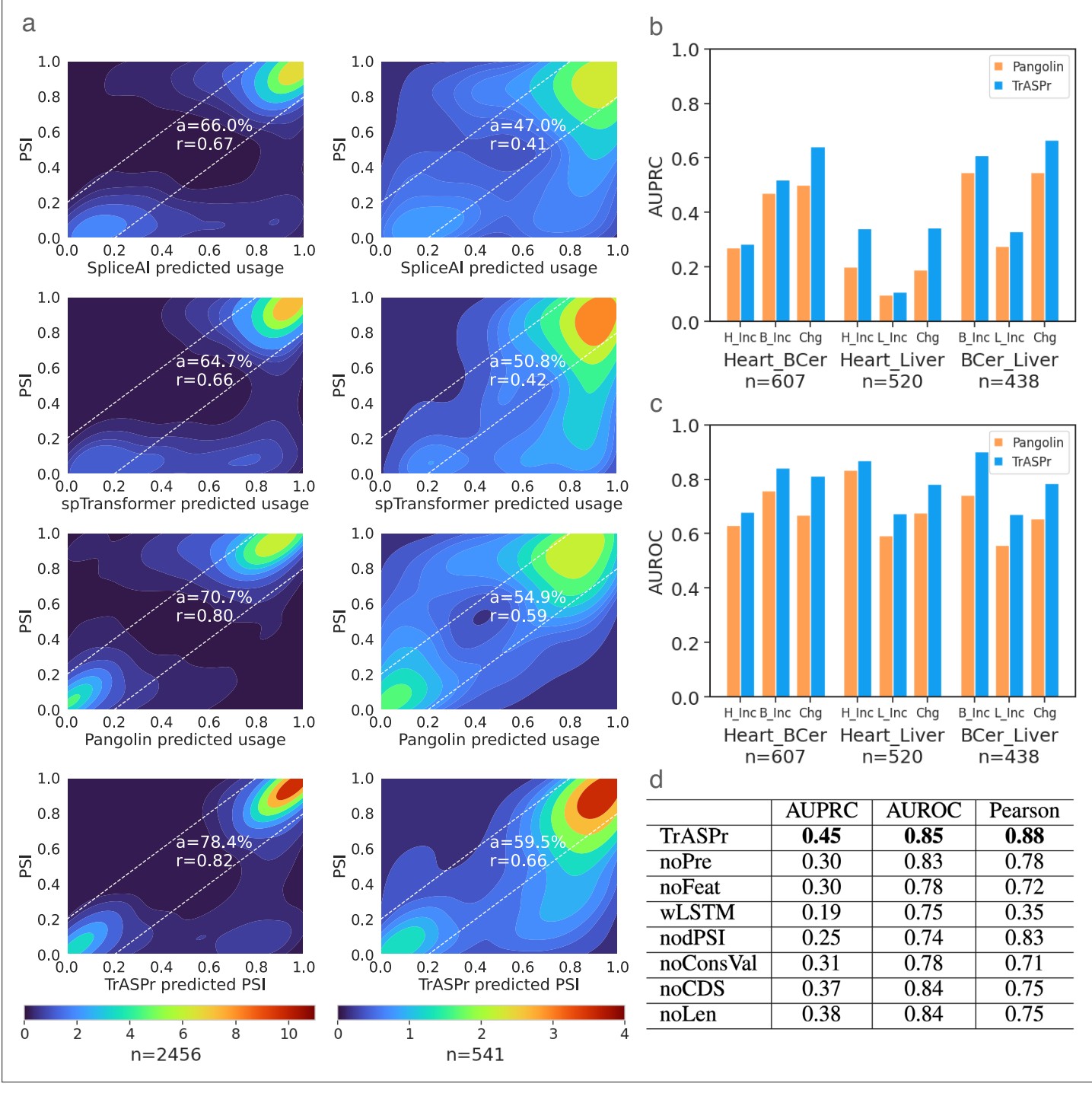

**Figure 2.** Comparison of PSI prediction results on GTEx dataset. (**a**) Heatmaps show the distribution of prediction vs. RNA-Seq values for all samples (left) and for samples involving exons that exhibit a change ($\Delta\Psi \geq 0.15$) between at least two tissues (right) for SpliceAI (top), SpliceTransformer (2nd row), Pangolin (3rd row), and TrASPr (bottom). $r$ is Pearson correlation, $a$ is the proportion of predictions approximately correct (within the dashed lines). $n$ is the number of samples (i.e. a cassette event measured in two tissues) in each setting, including heart-atrial appendage, cerebellum, and liver. (**b**) AUPRC for predicting events that are differentially included ($\Delta\Psi \geq 0.15$) in one tissue (e.g. heart) compared to another (e.g. liver) and for predicting differentially spliced events ($|\Delta\Psi \geq 0.15|$) between the two tissues. The tissue pair is denoted at the bottom. (**c**) Same as b above but for AUROC. Evaluations in a-c include samples from tissues Pangolin was originally trained on (heart-atrial appendage, cerebellum, and liver). (**d**) Ablation study across all pairs of six GTEx tissues (the above three as well as lung, spleen, and EBV-transformed lymphocytes). AUPRC and AUROC are averaged across all tissue pairs for the change vs no-change prediction task as in b-c, while Pearson correlation is for PSI as in a. Top row (TrASPr) is the full model with pre-trained transformers. noPre - same structure and input as TrASPr but trained from scratch. noFeat - same train/pretrain as TrASPr but without

*Figure 2 continued on next page*

*Figure 2 continued*

extra features. wLSTM - model with a bidirectional LSTM instead of Transformer and without the extra features. nodPSI - remove dPSIs in target function. noConsVal - same train/pretrain as TrASPr but without conservation value feature. noCDS: without coding region-related indicators and frame shifting features. noLen: without exon/intron length features.

The online version of this article includes the following figure supplement(s) for figure 2:

**Figure supplement 1.** GTEx test set Pearson correlation for each model, binned by the combined length of the cassette exon's upstream and downstream introns.

**Figure supplement 2.** Same as *Figure 2—figure supplement 1* but here test cases are binned by the alternative exon's length.

**Figure supplement 3.** Correlation between gene expression and SpliceTranformer's splice site usage.

**Figure supplement 4.** Tissue PSI values in GTEx samples (top - cerebellum, bottom - liver) vs. SpliceTransformer usage.

**Figure supplement 5.** Differential usage (x-axis) vs. differential splicing (dPSI, y-axis) for cassette exons in chromosomes 7, 8 assessed for the three GTEx tissue pairs used in the main text (*Figure 2*): heart_BCer, heart_liver, and BCer_liver.

(Pangolin) achieves $a = 0.71, r = 0.8$, but TrASPr outperforms both with improved PSI approximation $a = 0.78, r = 0.82$. Still, those statistics are dominated by extreme values, such that 33.2% are smaller than 0.15 and 56.0% are higher than 0.85. Furthermore, most cassette exons do not change between a given tissue pair (only 14.0% of the samples in the dataset, i.e. a cassette exon measured across two tissues, exhibit $|\Delta\Psi| \geq 0.15$). Thus, when we repeat this analysis only for samples involving exons that exhibited a change in inclusion ($|\Delta\Psi| \geq 0.15$) between at least two tissues, performance degrades for all three models, but the differences between them become more striking (*Figure 2a*, right column). The SpliceAI model is not tissue-specific and drops to $a = 0.47, r = 0.41$. As expected, the tissue-specific Pangolin model improves on SpliceAI to achieve $a = 0.55, r = 0.59$, but TrASPr outperforms both with $a = 0.60, r = 0.66$. Finally, the recently published SpliceTransformer performed similarly to SpliceAI and worse than Pangolin (see *Figure 2a*), and we therefore did not consider it any further.

One potential reason for the above differences in performances is data distribution shifts. Specifically, Pangolin, which exhibited the best performance compared to SpliceAI and SpliceTransformer, was trained on a different algorithm's output (SpliSER) and a different dataset. To assess if data distribution was the main reason for the performance gap, we first retrained Pangolin on MAJIQ's PSI and found no substantial differences in performance (Pearson correlation 0.78 and 0.59 for data shown in *Figure 2a* left and right column, respectively). Next, instead of using the pretrained Pangolin, we tried to adopt the model and retrained it from scratch on the same data as TrASPr. Specifically, since our model only trained on human data with six tissues at the same time, we modified Pangolin from the original 3' and 5' splice site usage predictions to 6 PSI outputs, centering the model's genomic window on either the 3' or 5' of the cassette exon. This test resulted in low performance (3' SS: Pearson 0.21, 5' SS: 0.26). This low performance may be due to various factors, such as the need for much more extensive data as part of this model's training, the need to include both 3' and 5' splice site recognition in the training procedure, or lack of optimal training parameters that we were not able to identify during re-training.

Another potential contributor to the differences we observed in performance between TrASPr and the other models is the length of the region spanned by the AS cassette event. As noted above, approximately 24% of the events in our data span windows greater than the 10 kb covered by Pangolin and SpliceAI, where the window is defined between the upstream exon's 5' splice site and the downstream exon's 3' splice site. Such large genomic windows where SpliceAI or Pangolin cannot 'see' those flanking splice sites might lead to degraded performance of those models on those events. To test this hypothesis, we divided our dataset into bins defined by the combined flanking introns length and reassessed performance for all four models. The results, shown in *Figure 2—figure supplement 2*, indicate TrASPr's overall improved performance is not due to such long AS events, as Pangolin's Pearson for events longer than 10 kb was slightly better than TrASPr. This somewhat surprising result may be due to different factors, such as a different splicing mechanism for cassette exons.flanked by long introns and additional information encoded by the long introns which is captured by Pangolin but not included in TrASPr input data. In contrast, when test cases are broken down by the cassette exon's length, TrASPr exhibits improved performance for longer exons (>200 bp) compared to the other methods (*Figure 2—figure supplement 1*).

The results above calibrate current SOTA performance for tissue-specific PSI prediction and raise a related question: Even if the exact PSI prediction is not accurate, can these models accurately predict which cassette exons are being regulated in a tissue-specific manner? Answering this question has been the focus of the original splicing code models (*Barash et al., 2010*; *Xiong et al., 2011*; *Barash et al., 2013*) and carries practical applications: Even if the magnitude of the change is not well calibrated, the models are likely able to identify regulatory features that are responsible for tissue-specific effects. In terms of assessing the models, this translates to measuring accuracy on three related *classification* tasks rather than regression tasks of differential exon inclusion, exclusion, and change vs no-change. Specifically, given two tissues $(c, c')$, differentially included exons in $c$ are exons for which $\Delta\Psi_{e,c,c'} = \Psi_{e,c} - \Psi_{e,c'} \geq 0.15$ while the negative set of exons in this classification task are exons for which $\Delta\Psi_{e,c,c'} < 0.05$. A similar definition can be made for differential inclusion in $c'$ (or equivalently, differential exclusion in $c$), while the task of change vs no-change is for detecting exons that exhibit clear splicing changes, regardless of the direction of change ($|\Delta\Psi_{e,c,c'}| \geq 0.15$), compared to those that do not ($|\Delta\Psi_{e,c,c'}| < 0.05$). *Figure 2b,c* summarizes the results on these three classification tasks in terms of AUPRC and AUROC. We note that for this task, SpliceAI is irrelevant as its predictions are not tissue-specific. The improvements offered by TrASPr for this task are particularly striking in AUPRC, which is arguably more important given the extreme label imbalance, with an average improvement of 1.28-fold and a maximum of 1.8-fold over Pangolin.

## Assessing TrASPr components

The improved performance by TrASPr naturally brings the question of which components of the model contributed to this improvement. We addressed this question through a series of ablation studies where we replaced the transformer with an LSTM (wLSTM), removed the additional features (noFeat), removed pre-training (noPre), and replaced the target function (nodPSI). The results, summarized in *Figure 2d*, indicate each of those components significantly improved performance. Finally, we also tried to use existing DNABERT, a larger BERT model trained on the entire human genome, instead of our lighter transformer pre-trained on splice junctions. We found DNABERT to be highly unstable for this task, requiring further parameter exploration, and even then, performance degraded. These results suggest that careful pre-training can be beneficial even when a smaller model is used and that since the coding sequence is only a small fraction of the human's DNA, DNABERT may be learning dependency structures that are less relevant for the task at hand.

## TrASPr generalizes to unseen cellular conditions and other AS types

Existing condition-specific splicing prediction models only generalize over unseen genomic sequences but not cellular conditions. This limitation implies that predictions were only performed for conditions for which training data already existed, limiting the usability of the splicing codes. To address this, we introduced a new component into TrASPr involving a Principal Component Analysis (PCA) that learns a latent space representation for the tissue or cellular condition (see Methods). As shown in *Figure 3a*, this latent space representation allows TrASPr to generalize from the six GTEx tissues to unseen conditions, including unseen GTEx tissues (*Figure 3a* mid row), and ENCODE cell lines (*Figure 3—figure supplement 1* mid row). Overall, using latent space representation for each tissue improves prediction accuracy compared to TrASPr lacking PCA (e.g. $a = 80.8\%$ vs $a = 79.9\%$ for GTEx tissues and $a = 78.8\%$ vs $a = 76.3\%$ for ENCODE cell lines), although naturally training on the additional GTEx and ENCODE conditions can lead to better performance (e.g. $a = 84.2\%$ for GTEx and $a = 83.4\%$ for ENCODE, *Figure 3a*, *Figure 3—figure supplement 1* top row). The differences become more clear again when focusing on samples involving exons exhibiting a significant splicing change ($|\Delta\Psi \geq 0.15|$) between at least two tissues (*Figure 3a* right column). Here, the six GTEx tissues model achieves a correlation of only $r = 0.39$ compared to $r = 0.53$ by the model with latent space embedding and slightly lower $r = 0.48$ for the eight tissue model. Encouraged by these results, we then tested the ability of TrASPr with a latent space representation for each tissue to predict differential splicing events in those previously unseen GTEx tissues. Notably, for this task, simply averaging predictions over training tissues cannot be used. The results, shown in *Figure 3b*, indicate that, as expected, performance is improved when the tissue data is available for training ('Token' model, with train samples involving eight tissues, including the two tissues used for test samples). However, using

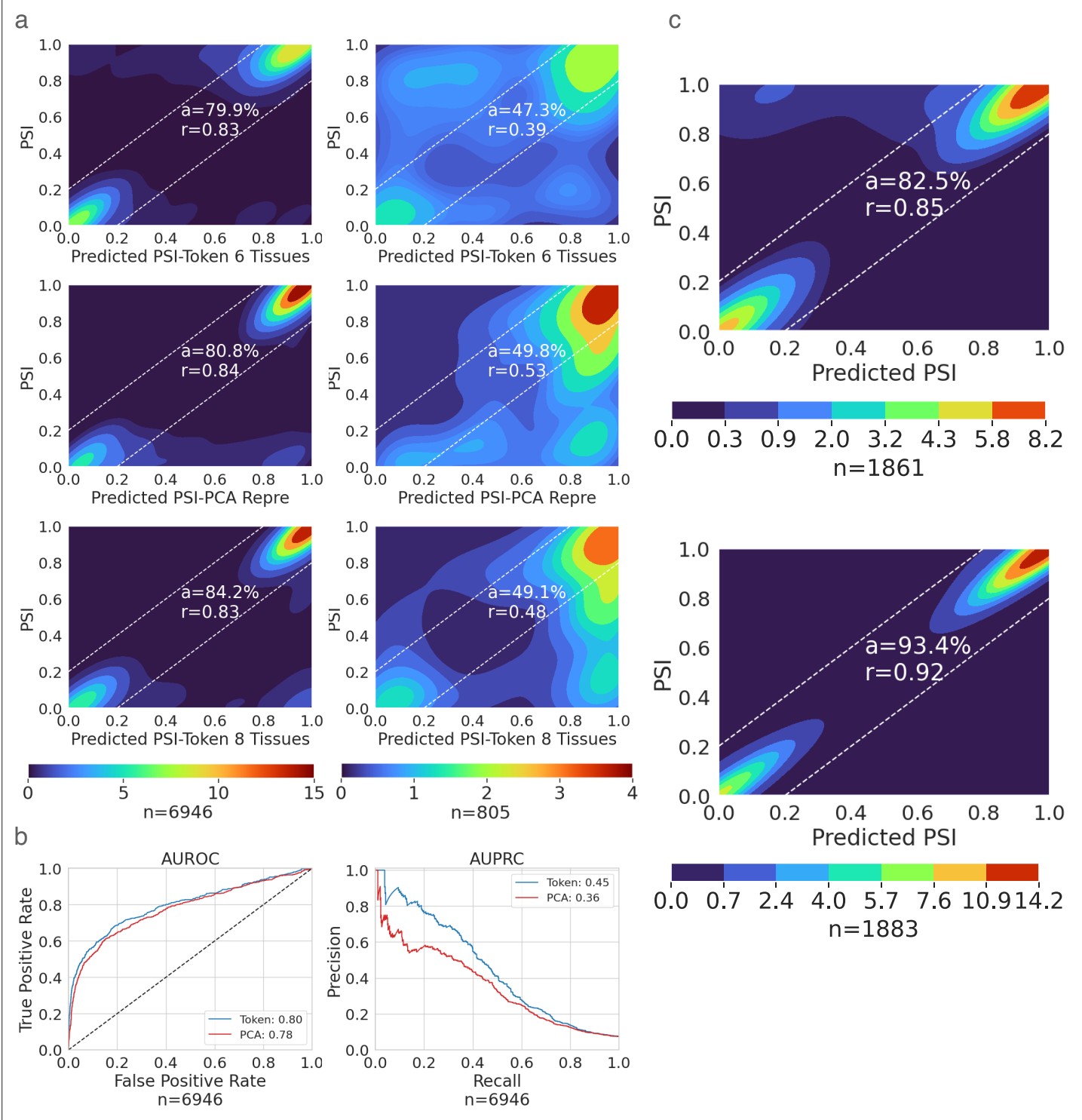

**Figure 3.** TrASPr prediction results in unseen conditions and alternative splice sites. *n* is the number of samples in each setting. (**a**) Performance of TrASPr on two GTEx test tissues (cortex and adrenal gland). Top: The tissues were first represented as tokens, and new tissue results were predicted based on the average over conditions during training. Mid: TrASPr used the PCA learned representation to predict AS in the two GTEx test tissues it never trained on. Bottom: TrASPr was trained on all eight tissues using token-based tissue representations and tested on the two GTEx test tissues. The left column includes all samples and the right one only has changing event samples. Changing events have inclusion level change larger or equal to 0.15 in at least one tissue pair. (**b**) AUROC and AUPRC plots for predicting change vs no-change events in the two GTEx test tissues used in (**a**), compared to the six original tissues. Blue: TrASPr with a token per tissue, trained on samples from all eight tissues. Red: TrASPr using PCA embedding to represent

*Figure 3 continued on next page*

*Figure 3 continued*

tissues, where samples from the two test tissues were not included in the training. (**c**) Prediction accuracy of TrASPr when applied to alternative 3′ (top) and alternative 5′ (bottom) splice sites.

The online version of this article includes the following figure supplement(s) for figure 3:

**Figure supplement 1.** TrASPr was trained on GTEx 6 tissues or GTEx +two ENCODE cell lines and then tested on two cell lines in ENCODE (HepG2 and K562).

TrASPr with a PCA latent space embedding to represent tissue type can still generalize quite well to predict differential splicing in the two unseen tissues.

Next, we wanted to assess the ability of our framework to generalize to AS events which are not exon skipping. We used the same pre-trained BERT models and trained TrASPr on the same GTEx samples, but quantified junction inclusion ($\Psi$) for alternative 3′ and alternative 5′ events as shown in *Figure 1a* instead of cassette exons. We observed similar performance to cassette exons for TrASPr (3′ $r = 0.85$, 5′ $r = 0.92$, see *Figure 3c*). In contrast, we observed significantly worse results for Pangolin and SpliceAI on this task (Pangolin: $r = 0.36, 0.31$, SpliceAI: $r = 0.13, 0.10$ for 5′ and 3′ splice sites dataset respectively). It is unclear why Pangolin's and SpliceAI's performance was much worse on these tasks. Of note, to the best of our knowledge, both models were not previously evaluated on these tasks. Thus, contributing factors may be the models' focus on cryptic splice site creation (by training on splice site identification as a binary yes/no task), exon skipping being more common, or that slight changes in splice site usage between two adjacent positions are not captured well by these models. Regardless, TrASPr results demonstrate that it can generalize to both unseen cellular conditions and non-cassette exon AS events as well.

## Predicting the effect of regulatory elements and mutations

Since TrASPr predicts PSI and dPSI directly from genomic sequences combined with related features, it should be able to capture the effect of both tissue-agnostic and tissue-specific regulatory elements as well as the effect of genetic mutations on those. To assess TrASPr's ability in such tasks, we first tested the effect of enhancing or degrading the core spliceosome 3′ and 5′ splice sites (see Data for details). *Figure 4a* shows the results of this analysis for lowly included alternative exons whose splice sites were enhanced (left) and highly included exons whose splice sites were weakened (right). In both cases, the observed effect on PSI predictions is as expected, causing an increase (left) or decrease (right) in inclusion level. Notably, the magnitude of the change can vary greatly, and in general, weakening the splice sites tends to have a stronger effect. This result is to be expected as there are other elements (e.g. branch point, polypyrimidine tract) that can affect splicing, so weakening strong splice sites is likely to have a strong effect, but improving splice sites may not be sufficient to create strong exon inclusion.

Next, we reasoned that given the model's structure and the known mechanisms involved in splicing regulation, the prediction accuracy should have a strong dependency on the relative position of the learned features. To test this hypothesis, we swapped the order of the two 3′ splice site transformers and the two 5′ splice site transformers around the cassette exon during test time. As expected, all these pairwise swaps significantly degraded performance, with Pearson correlation on the same data used in *Figure 2a* dropping from 0.88 to 0.81 and 0.77 when the two 3′ or 5′ splice site transformers were swapped. Swapping the 3′ and 5′ transformers around the alternative exon abolished predictive power (Pearson correlation 0.17).

The above analysis serves as a qualitative sanity check that the model captures core splicing signals. However, a large-scale mutagenesis experiment can provide a more thorough quantitative analysis. For this, we used 6106 sequence samples where each sample may have multiple positions mutated (i.e. mutation combinations) in exon 2 of CD19 and its flanking introns and exons *Cortés-López et al., 2022*. The total number of mutations hitting each of the 1198 genomic positions across the 6106 sequences is shown in *Figure 4b* (left), while the distribution of effects (|$\Delta\Psi$|) observed across those 6106 samples is shown in *Figure 4b* (right). To this data, we applied three testing schemes. The first is a standard fivefold CV where 20% of combinations of point mutations were hidden in every fold while the second test involved 'unseen mutation' (UM) where we hide any sample that includes mutations in specific positions for a total of 1480 test samples. As illustrated by the CDF in *Figure 4b*, most samples (each sample may involve multiple positions mutated) do not involve significant splicing

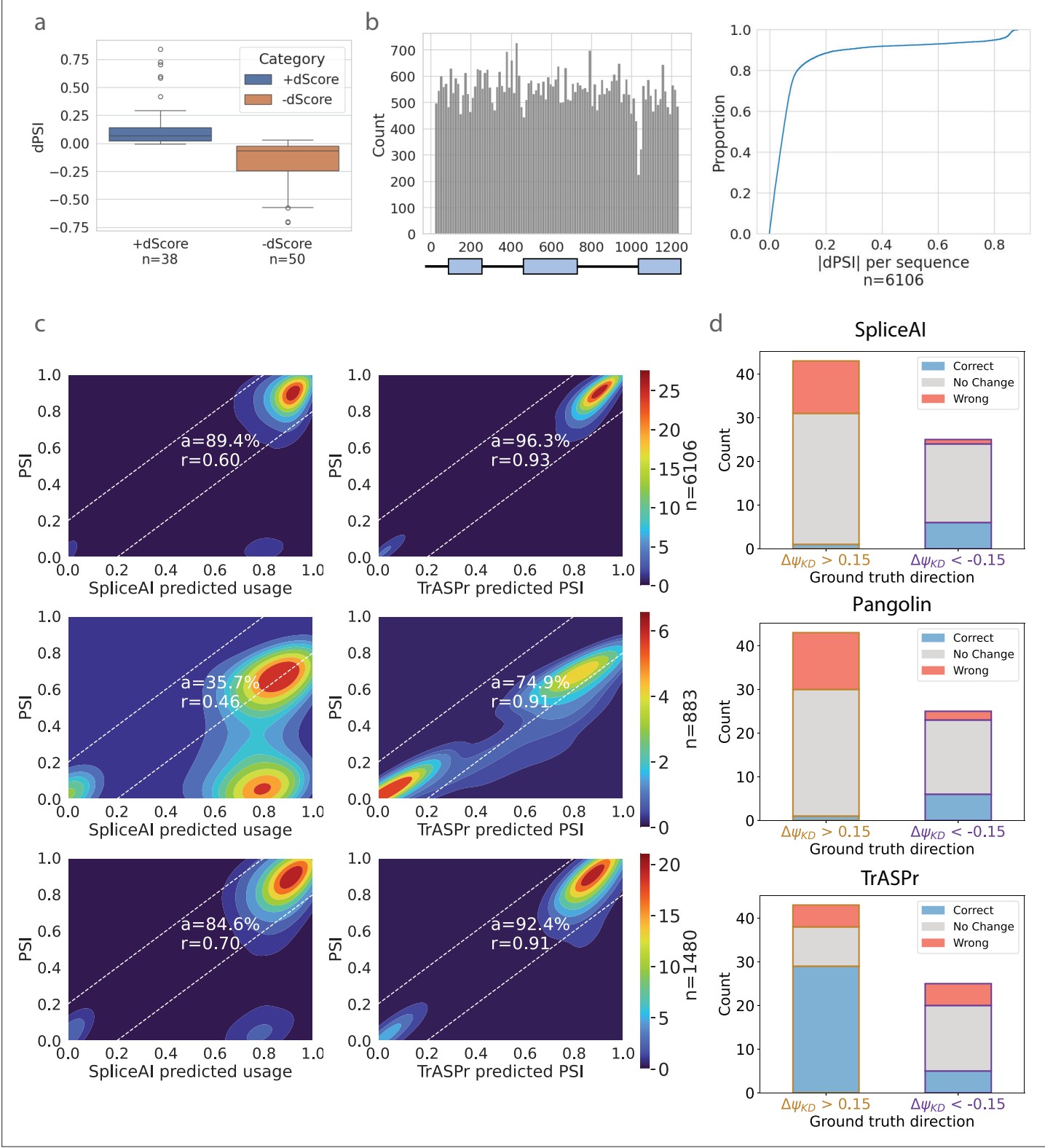

**Figure 4.** TrASPr prediction results on mutation effect. (**a**) Whisker plot for splice site mutation effect on predicted PSI when weak splice sites are made strong (blue, left) and when strong splice sites are made weak (brown, right). (**b**) Distribution of mutation positions in CD19 dataset (left) and the CDF of the marginal effect per each of those sequences (right). (**c**) Heatmaps showing the performance of SpliceAI (left column) and TrASPr (right column) in predicting the effect of mutations shown in b, under three settings: random fivefold cross-validation (top row), random fivefold cross-validation for changing mutations only (middle row), and single unseen mutation filter (bottom row). *n* indicates the number of cases in the test set. (**d**) Predicting the

*Figure 4 continued on next page*

*Figure 4 continued*

effect (dPSI direction) of RBPs KD by mutating their corresponding sequence motifs. Blue, grey, and red correspond to correct, no change, and opposite direction prediction, respectively.

The online version of this article includes the following figure supplement(s) for figure 4:

**Figure supplement 1.** Heatmaps showing the performance of Pangolin in predicting the effect of mutations for CD19 dataset with the same settings as shown in *Figure 4c*.

**Figure supplement 2.** CDF for the difference between TrASPr test data predictions (PSI') and ground truth PSI derived from MAJIQ on ENCODE dataset.

**Figure supplement 3.** Predicting the effect of tissue specific splice factors exhibits strong positional biases.

changes. Thus, we also performed a third test using only the 883 samples where mutations cause significant changes ($|\Delta\Psi| \geq 0.15$). The results of these three testing schemes are shown in *Figure 4c*. As expected, TrASPr performance degrades as the testing becomes more strict, yet it significantly outperforms SpliceAI in all settings. As for Pangolin, while out of the box performance is similar to SpliceAI, retraining Pangolin on this data results in performance closer to TrASPr (see *Figure 4— figure supplement 1*).

A caveat of the previous evaluation is that it is not focused on RBPs and tissue-specific splicing regulatory elements, with strongly affecting mutations concentrated around splice sites. To assess TrASPr predictions for RBP regulatory elements, we turned to ENCODE data. First, we assessed whether TrASPr is able to predict exon inclusion in those new conditions accurately. As shown in *Figure 4—figure supplement 2*, TrASPr predicts $\Psi$ within a 10% accuracy in almost 90% of the test cases, indicating excellent accuracy for the ENCODE cell lines. Next, we used RBP KD data (*Maurin et al., 2023*; *Van Nostrand et al., 2020*) to compile a list of 68 cassette exons regulated by four well-studied, condition-specific RBPs (TIA1, PTBP1, QKI, RBFOX). For each of these targets, we recorded the experimentally measured effect (increased inclusion or exclusion) of the matching RBP KD and compared it to the predicted effect by the model. Model prediction for the RBP KD effect was mimicked by mutating the genomic sequence corresponding to the RBP binding motif. Since most short sequence mutations are not expected to significantly alter PSI unless these 'hit' a core splicing signal, we set predictions to 'no change' when the dPSI effect was below the 95th percentile of effects observed by random mutations (see Data for details). Overall, TrASPr performed well on most positive effect cases but predicted around half of negative effects as no change. The correlation coefficient for the dPSI effects was 0.34, and the fraction of correctly called changes was over 50%, while only 20% were called incorrectly. In comparison, SpliceAI and Pangolin performed significantly worse (*Figure 4d*), with both predicting correctly only for 10.3% of the cases and predicting no effect for 70.6% and 67.6%, respectively. These results indicate both SpliceAI and Pangolin struggle to capture condition-specific regulatory elements.

Finally, we decided to test whether TrASPr learns positional preferences for tissue specific splice factors. For this, we took 40 cassette exons exhibiting high inclusion (PSI >0.85) in our GTEx tissue set, and 40 cassette exons exhibiting low inclusion (PSI <0.15). We then performed in silico mutagenesis analysis by inserting either a FOX binding motif (TGCATG) or a QKI motif (ACTAAC) in each possible position along the cassette exon, comparing the effect to 10 randomly chosen sequences (see Methods). The results of this analysis, shown in *Figure 4—figure supplement 3*, recapitulate known binding maps for both splice factors with a strong preference for areas less than 150 up or downstream of the alternative exon (*Hall et al., 2013*; *Jangi et al., 2014*; *Van Nostrand et al., 2020*). Inline with the FOX motif map (*Jangi et al., 2014*; *Gazzara et al., 2017*), inclusion was most strongly promoted by FOX when the binding site was downstream of the alternative exon, and upstream when promoting exclusion. However, the strongest QKI inclusion effect occurred when the binding site was introduced up rather than downstream, contrary to what its RNA motif map would suggest (*Hall et al., 2013*). Such effects have been reported in the literature and may be the result of context-specific effects accounted for by the model, such as additional RBPs binding in the region, but may also simply be inaccuracies of the model in terms of predicting the direction of effect in those cases.

## TrASPr enables identification of new tissue-specific splicing changes and regulatory elements

Given the strong performance we observed in predicting tissue-specific splicing changes, we wanted to test whether TrASPr can be used to predict previously unknown splicing changes and regulatory elements. To do so, we employed LSV-Seq (*Yang et al., 2024*), a recently developed targeted sequencing method from our lab which allows for enrichment and quantification of splicing events. We hypothesized that we could use TrASPr to recover tissue-specific splicing changes from previously detectable but unquantifiable low-coverage splicing events identified from analysis of GTEx RNA-Seq experiments. We first created a list of such cassette events, then assessed those for tissue-specific splicing ($\Delta\Psi > 0.1$) using TrASPr. Targeting the 787 predicted tissue-specific cassette exons predicted with LSV-Seq resulted in 558 events that had sufficient coverage (> 30 reads across at least 2/3 tissues) to be confidently quantified as changing ($\Delta\Psi > 0.1$) or nonchanging ($\Delta\Psi < 0.05$) compared to TrASPr predictions. Overall, compared to an expected success rate of 4.7% for random exon selection, TrASPr target-selection achieved good validation rates ranging from 48.8% to 55.8% validation rate depending on the stringency of the threshold for the predicted $\Delta\Psi$ (0.1, 0.15, or 0.2, corresponding to 'passed', 'stringent', and 'very stringent' shown in *Figure 5a*).

Overall, the above analysis led to the identification of 169 new tissue-specific cassette exons, two of which are highlighted in *Figure 5b*. The first example is of a brain cerebellum-specific cassette exon skipping event predicted by TrASPr in the ATP13A2 gene (aka PARK9). ATP13A2 is a lysosomal transmembrane cation transporter, for which loss-of-function mutation has been linked to early-onset of Parkinson's disease (PD; *Dehay et al., 2012*; *Ramirez et al., 2006*; *Zhang et al., 2022*). Here, we detect an exon in a cytosolic loop of the protein with elevated skipping in the cerebellum. The major protein isoform with this exon skipped, which also contains variations in the C-terminal, is degraded by the proteasome after mis-localization to the endoplasmic reticulum (ER) membrane (*Ugolino et al., 2011*). However, the specific function of this exon in the protein remains unknown. Interestingly, many PD-associated mutations degrade ATP13A2 through a similar mechanism (*Podhajska et al., 2012*; *Ramirez et al., 2006*; *Ugolino et al., 2011*). But while ATP13A2 proteasomal degradation can have significant consequences for disease, it is unclear what role this process or its regulation by AS plays in normal tissue.

A second example of validating cerebellum-specific cassette exon skipping predicted by TrASPr involves the PTPN23 gene. PTPN23 is an essential gene with diverse molecular functions, including degradation of ubiquitinated proteins (*Doyotte et al., 2008*; *Ma et al., 2015*), regulation of the SMN complex (*Husedzinovic et al., 2015*), and regulation of neuron pruning (*Loncle et al., 2015*). Proper regulation of PTPN23 expression is important, and its dysregulation often causes disease in many tissues (*Gingras et al., 2009*). PTPN23 haploinsufficiency or under-expression is associated with neurodevelopmental delays (*Bend et al., 2020*), heart defects associated with cardiomyopathy (*Xu et al., 2024*), and cancer growth (*Cao et al., 1998*; *Manteghi et al., 2016*; *Singh et al., 2023*). Here, TrASPr predicts tissue-specific splicing of an exon in PTPN23, whose skipping leads to a premature termination codon and nonsense-mediated decay of PTPN23. This suggests a mechanism for tissue-specific regulation of PTPN23 expression and could also be potentially used for therapeutic modulation of PTPN23 expression in disease.

Next, we turned to experimentally test specific regulatory elements. For this, we first trained TrASPr to predict differential splicing between two cell lines, HEK293T (embryonic kidney cell line) and SH-SY5Y (neuroblastoma cell line), and then selected specific positive (affecting splicing) and negative (non-affection) regions around the cassette exon for validation (see Methods). The results for validating such regions around two cassette exons are shown in *Figure 5c,d* where g1-5 are the sgRNAs used to target dCas13 region, and Mut1, 2… are the mutations selected by TrASPr based on dPSI. For all sites tested, we observed a strong agreement with the computational predictions. Taken together, these results point to the usefulness of TrASPr in predicting tissue-specific splicing and regulatory elements that have not been observed or detected before. This ability raises the question of whether we can utilize TrASPr to then design new sequences as well. We turn to tackle this question next.

## Assessing BOS sequence generation

In order to generate genomic sequence with a specific splicing profile, we use TrASPr to define a black-box objective function and optimize it using BO. Specifically, using BO, we optimize over the

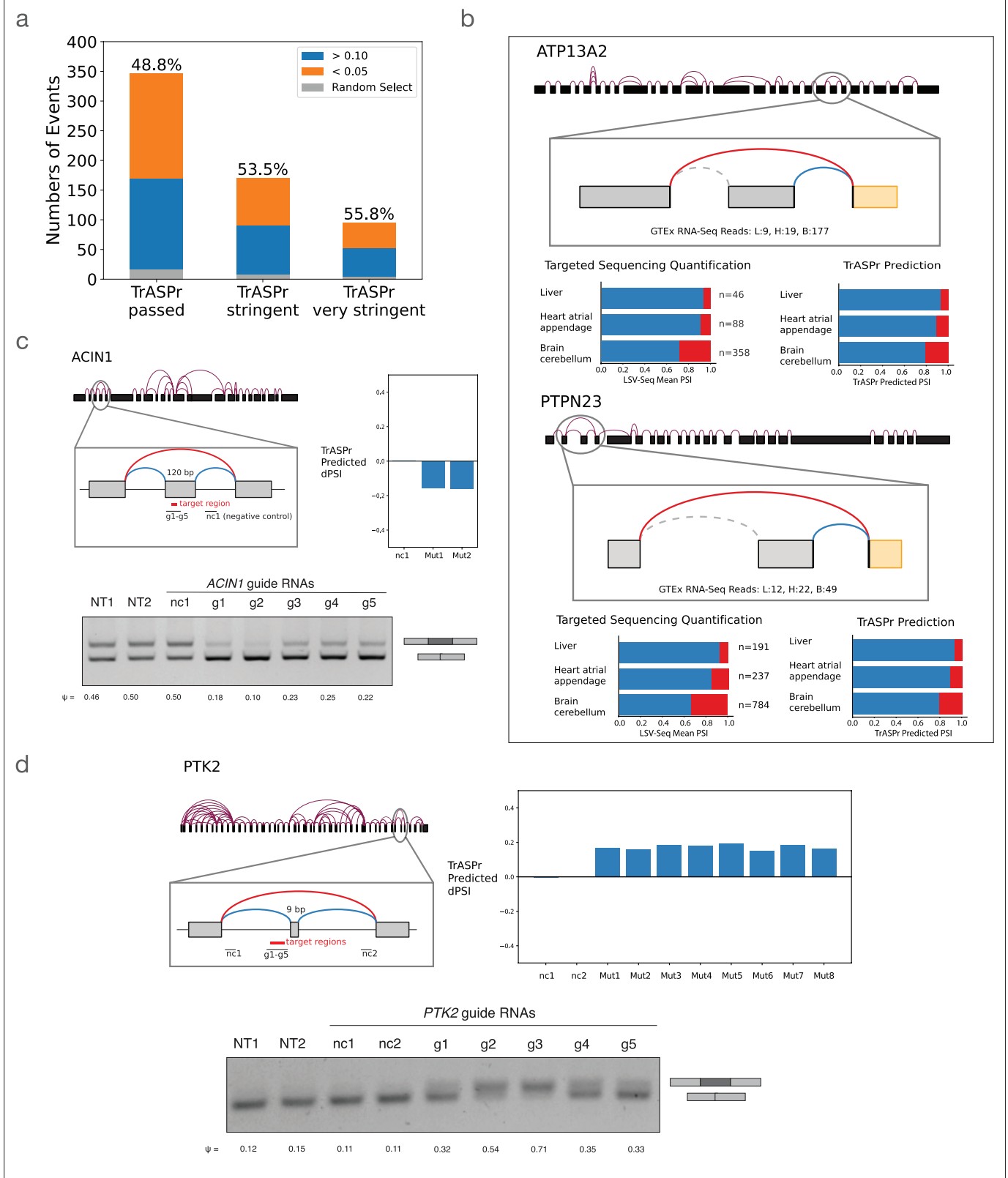

**Figure 5.** Experimental validations for TrASPr predictions. (**a**) Bar plot for the validation rate of low coverage AS events predicted by TrASPr to exhibit tissue-specific splicing between cerebellum, liver, and heart-atrial appendage. Validation rate was between 48.8% to 55.8%, depending on the prediction stringency, discovering a total of 169 new tissue specific events. (**b**) Two examples of newly found tissue-specific AS events from (**a**). For each case, the top graph illustrates the splicing context of the event. Two bar plots show the comparison between LSV-seq experimental results (bottom left)

*Figure 5 continued on next page*

*Figure 5 continued*

and TrASPr predictions (bottom right). (**c,d**) Two AS events where specific regions were targeted by dCas13d, including elements predicted by TrASPr to have significant regulatory effect and negative control regions. The bar plot (top right) shows the predicted inclusion level changes by TrASPr for 6b long windows in the tested region. Effects of dCas13d targeting were assessed by RT-PCR (bottom, NT = non-targeting, nc = negative control).

The online version of this article includes the following source data for figure 5:

**Source data 1.** Original files for RT-PCRs displayed in *Figure 5c* (left), d (right).

**Source data 2.** Original RT-PCR images corresponding to *Figure 5*, panel c (top) and panel d (bottom).

space of possible RNA sequences to find sequences that successfully accomplish some task according to TrASPr (e.g. increase exon inclusion), which serves as a teacher or an oracle. BO is an iterative, model-based optimization procedure involving the use of a surrogate model (typically a Gaussian process [GP]) to approximate the objective function and iteratively select the most promising candidates to evaluate. In our case, we use a GP model to iteratively select candidate RNA sequences, which are then evaluated using TrASPr. Note that GPs and other standard surrogate models cannot be defined directly over the structured, combinatorial space of all possible RNA sequences. We therefore use a special kind of BO called latent space Bayesian optimization (LSBO). LSBO allows BO to be applied over structured search spaces by first learning a continuous, numerical, latent space representation of the structured space. To obtain a latent space representation of the RNA sequence space, we pre-train an (unsupervised) transformer-based variational auto encoder (VAE) model on a large set of RNA sequence data. We then define our GP surrogate model over the learned latent space and run LSBO. We refer to this method as BOS (see Methods for more details).

We assess the ability of BOS to generate RNA sequences for a variety of tasks and compare it to two baselines. The first baseline method randomly mutated 3, 6, 15, and 30-mers in different regions in the hope of achieving the desired effect. The second baseline is a genetic algorithm (GA) as in *Sample et al., 2019*, originally applied to design 5' UTR sequences. In all cases, we assessed how many of the generated sequences matched the desired criteria (e.g. increase exon inclusion) and what was the best scoring sequence found (e.g. max dPSI achieved). Note that in these evaluations we assume the values predicted by TrASPr (PSI, dPSI) for a generated sequence are correct and only assess each algorithm's ability to generate candidate sequences efficiently.

The first task we used to assess BOS was to improve inclusion levels (PSI) of weak cassette exons. Here, we started from cassette exons where either the 3' or 5' splice site was weakened and instructed BOS to improve inclusion levels. BOS was able to achieve a mean success rate of over 50% (*Figure 6a*) and a mean increase of inclusion of 40%. In comparison, both RM and GA achieved less than half of BOS's success rate (~21%) with lower inclusion levels. Indeed, when we used the MaxEnt (*Yeo and Burge, 2003*) algorithm to score the splice sites in the generated sequences, we find BOS is able to produce sequences with significantly better splice sites (*Figure 6a* bottom).

While the above results indicated strong performance by BOS, they relied on computationally assessing performance by TrASPr and the MaxEnt algorithm. To assess BOS generated sequences with respect to experimental evidence, we employed the high-throughput mutagenesis CD19 exon 2 experiment (*Cortés-López et al., 2022*). Overall, this experiment included mutations to 1198 positions spanning the entire cassette exon region as shown in *Figure 4b*. To assess site selection by BOS, we computed the marginal effect observed per position when mutated and compared it against BOS's frequency of selecting it. Plotting the location of mutated positions to decrease exon2 inclusion, we see BOS 'locks' on the two areas proximal to the splice sites which have the largest marginal effect (*Figure 6b*). More generally, we find that BOS learns to favor more affecting positions in terms of their marginal dPSI effect, selecting the top 10% of affecting positions (marginal dPSI >0.114) 31.2% compared to only 11.0% for positions at the bottom 10% (dPSI <0.03, binomial p-value $< 10^{-4}$).

Next, to assess the ability to alter splicing events in a tissue-specific manner, we supplied each algorithm with lowly included cassette exons and requested no more than 30 edits that will make those have a relatively increased inclusion in cerebellum compared to other tissues of at least 30%. On this task, we considered generated sequences with high inclusion levels in cerebellum (PSI>0.5) and low inclusion in other tissues (PSI<0.2) as successful. Notably, this is a much harder task compared to the non-tissue specific ones described before. Indeed, when assessing 10 different lowly included exons as a starting point, TrASPr was only able to match the user constraints 20% of the time on

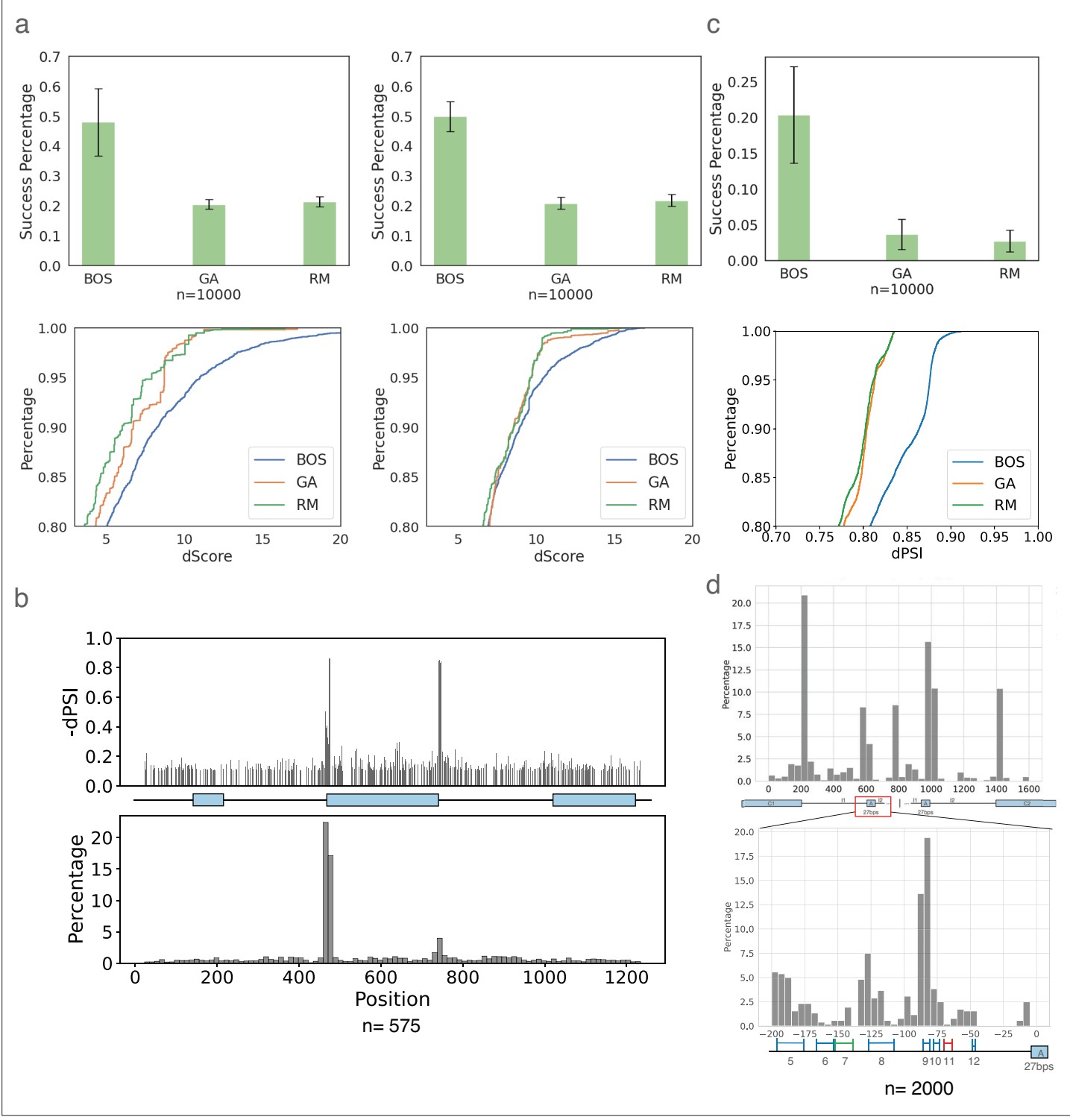

**Figure 6.** RNA design results by BOS. (**a**) Results for the task of improving inclusion of weak cassette exons (n=8 exons). Top: Bar plots for success rate in achieving the desired design task (increased inclusion). Error bars represent standard deviation over the set of exons tested. Bottom: CDFs over the best designed sequences (top 20%) by the MaxEnt splice site score change between the original sequence and proposed sequence. GA, genetic algorithm; RM, random. (**b**) BOS generation results for CD19 mutation dataset, showing 575 generated sequences across 4846 possible positions. The positions mutated by BOS (bottom) capture regions close to the alternative exon splice sites whose mutations have strong marginal effects on inclusion levels (top). (**c**) Comparison of BOS, GA, and RM on tissue-specific (cerebellum) sequence generation. Different start sequences (n=10) are randomly chosen from cassette exons exhibiting low inclusion levels. Every algorithm is tasked with adopting the start sequence to achieve cerebellum-specific

*Figure 6 continued on next page*

*Figure 6 continued*

high-inclusion ($\Psi \geq 0.5$ for cerebellum, otherwise $\Psi \leq 0.2$) within 30 edits. Top: Success rate for this task. Bottom: The achieved improvement (dPSI) for the top 20% sequences generated by each algorithm. (**d**) BOS generation results for neuronal specific Daam1 exon 16. Bar plots indicate the distribution of 27206 hits where BOS mutated from 2000 sequences. The bottom plot is the zoom-in region of the top one. Regions that were validated experimentally by mutating them in a mini-gene system are marked either blue (yes) or red (no) depending on if TrASPr that teaches BOS is able to predict the effect of those segments. The green region indicates a region that doesn't affect the inclusion level and is predicted correctly by TrASPr.

The online version of this article includes the following figure supplement(s) for figure 6:

**Figure supplement 1.** BOS sequence edits for neuronal specific splicing.

**Figure supplement 2.** Generated sequences result for Daam1 gene exon 16 (n=4392).

average, but GA and RM exhibited a much lower success rate of approximately 4% (*Figure 6c* top). The differences in performance between the algorithms were even more striking when considering the best 20% of proposed sequences (*Figure 6c* bottom). Finally, when we assessed the contribution of intronic mutations (more than 8 bp away from the nearest splice site) introduced by BOS, we found these contributed most of the neuronal specific signal, with more pronounced effects achieved by the intronic upstream mutations (*Figure 6—figure supplement 1*).

Finally, to assess BOS generation with respect to experimentally tested tissue-specific regulation, we gave BOS the sequences proximal to exon 16 in the Daam1 gene and instructed it to reduce inclusion levels but without abolishing exon inclusion (PSI >0.1). Exon 16 is a well-studied, neuronal specific, micro-exon which has been shown to have high inclusion levels in cerebellum and in N2A cell lines ($\Psi = 0.66$). As such, previous work using a mini-gene reporter assay in N2A cells mapped several regulatory elements in the upstream intron that affect its inclusion (*Figure 6d*, colored regions). On this task, the best random mutation setting (30-mers) successfully generated 177 out of 4392 sequences (4.03%) that reduced inclusion by more than 0.2. The GA successfully generated 210 sequences (4.7%), while BOS generated 1331 successful sequences (30.3%) that matched the constraint of dPSI > 0.2 (See *Figure 6—figure supplement 2*). Furthermore, BOS also significantly outperformed the two baselines in terms of the best candidate sequence generated (0.71 maximum dPSI, compared to 0.61 and 0.53 for the GA and the random sampling algorithms, respectively). The majority of positions BOS chose to edit were, as expected, around the splice sites (*Figure 6d* top). However, zooming in on the upstream intron (*Figure 6d* bottom), we found BOS repeatedly mutated the validated enhancing regulatory elements, avoiding the negative control region (green). As expected, BOS still failed to suggest editing the red region on which TrASPr itself fails, demonstrating that the generative process is inherently limited by the capabilities of the oracle.

Overall, our analysis of in silico predictions and experimental assays indicates that BOS is able to efficiently capture regulatory elements in a given sequence, including both splice site signals as well as deep intronic elements, then capitalize on those to generate sequences matching a given splicing target function.

## Discussion

In this study, we offer two main contributions. First, we propose a new tissue-specific splicing code model, TrASPr. TrASPr's architecture leverages the transformer attention mechanism while utilizing multiple transformers, each focused on specific genomic regions. This design ensures the model concentrates on areas most relevant to splicing regulation without requiring excessively large models and training datasets.

We demonstrated that TrASPr significantly outperforms current state-of-the-art models in predicting PSI and dPSI across multiple datasets, even when those models used considerably larger genomic windows. Moreover, to our knowledge, this is the first demonstration of the ability to predict PSI and dPSI under previously unseen experimental conditions and to predict variations in both 3' and 5' splice sites. Using TrASPr, we generated predictions that were experimentally validated to identify tissue-specific splicing variations undetectable by previous RNA-Seq, as well as condition-specific regulatory elements.

In terms of related work, the models closest to TrASPr are Pangolin (*Zeng and Li, 2022*) and the recent SpliceTransformer (*You et al., 2024*), both designed to predict tissue-specific splicing. SpliceTransformer employs a similar approach to SpliceAI and Pangolin, scoring genomic positions

as 3' or 5' splice sites within a window spanning 4,000 nucleotides upstream and downstream. Additionally, SpliceTransformer incorporates what the authors define as splice site 'usage' across different GTEx tissues. Adding this usage statistic into its target function is designed to aid in tissue-specific splicing prediction. However, in practice, we found that SpliceTransformer performed similarly to SpliceAI and worse than Pangolin (see *Figure 2a*). A potential reason for this may lie in the SpliceTransformer's 'usage' based target function. 'Usage' is defined in *You et al., 2024* as the fraction of GTEx tissue samples in which a splice site was identified. For instance, a value of 0.1 indicates the splice site was detected (i.e., supported by more than one read) in 10% of the tissue samples. Consequently, this addition in the SpliceTransformer's model to SpliceAI's original target function correlates poorly with splicing or differential splicing quantification. Instead, 'usage' primarily reflects splice site detection capability, which is largely influenced by read depth and gene expression levels (see *Figure 2—figure supplement 3*). We note that these results do not negate SpliceTransformer's ability to detect mutations that disrupt or create splice sites for downstream analysis, similar to SpliceAI and Pangolin.

The second key contribution of this study is the formulation of RNA sequence design with specific splicing characteristics as a BO problem. We developed the BOS algorithm, which uses TrASPr as an oracle to tackle this design challenge by introducing biologically plausible mutations. Across various tasks, we showed that BOS effectively generates RNA sequences with the desired splicing changes, incorporating mutations that selectively create or disrupt core splicing signals and intronic regulatory elements as required. Here, we compared BOS to a previously used genetic algorithm and randomly introduced mutations as a baseline. However, more recent publications proposed additional methods to generate genomic sequences with certain characteristics. For example, *Wilkins et al., 2024* used SpliceAI predictions to create cryptic splice sites, *Schreiber et al., 2025* applied greedy editing of a given sequence by treating each position as a distribution over four nucleotides and computing the gradient per position, and *Biancalani et al., 2024* developed generative flow networks to edit or generate sequences with desired properties. Although these works did not focus on designing sequences for the tissue-specific splicing as presented here, such algorithms can be potentially adopted for this task in future works.

There are several potential applications for the work proposed here. First, TrASPr provides a relatively 'lightweight' LLM that can be easily fine-tuned for additional cellular conditions of interest. This capability enables the detection of condition-specific splicing and associated regulatory elements in scenarios where experiments have not yet been conducted or in genes with low coverage. It can also be used to assess the effects of genetic variants, such as resolving undiagnosed cases of rare diseases. This application was recently demonstrated in *Wagner et al., 2023*, where SpliceAI emerged as a top-performing model. For RNA design tasks, BOS and similar algorithms offer valuable tools for synthetic biology studies and therapeutic applications. For example, these algorithms could guide the design of sequences to target with ASO (antisense oligonucleotide) therapies or prime editing approaches.

While the above applications are exciting, we acknowledge several limitations and areas for potential improvement in this work. First, although TrASPr demonstrated significant advancements in PSI and dPSI predictions, it remains far from perfect. Specifically, the current model was not optimized for predicting the effects of genetic variations and can only capture mutations within 200 bases of existing splice sites. Additionally, we have not evaluated TrASPr on complex splicing variations involving multiple alternative splice junctions. However, we note that such events are often simplified when analyzing changes between two cellular conditions (*Vaquero-Garcia et al., 2023*).

As for the analyses conducted here, it is important to recognize that the labels used for evaluating prediction tasks are inherently noisy and limited in number. For instance, RNA-Seq quantification is prone to noise, as are RBP binding assays such as eCLIP. Furthermore, RBP regulatory motifs are relatively crude representations, meaning many targets may be missed. Changes observed upon RBP KD could also arise from indirect effects, such as another RBP influenced by the KD, or from other sequence motifs.

Finally, we note a recent trend in AI for genomics toward the development of large 'foundation' models that can be fine-tuned for specific tasks. At this time, it remains unclear in what tasks such models can deliver superior performance or when lighter, task-specific models may be more accurate or useful. A related question is how easily a model can be adapted to handle new tasks, conditions, or datasets (*Penzar et al., 2023*).

In this work, we present a 'middle ground': leveraging the framework of LLM pre-training while constructing models tailored to specific tasks. This approach results in significantly lighter models that require fewer computational resources and are easier to adapt to new tasks. Another related approach is to create smaller, dedicated models that adopt a foundation model latent representation to specific tasks (e.g. *Daoud and Ben-Hur, 2025*; *Singh et al., 2025*). We envision TrASPr, paired with BOS for design tasks, as a 'base' model that can be readily fine-tuned for specific cell types, tissues, RBP KDs, and other conditions. Indeed, we demonstrated TrASPr's ability to generalize to alternative 3' and 5' splice site usage and to unseen cellular conditions by utilizing PCA for a latent space representation of cellular conditions.

In summation, we are excited for the future of LLM for RNA prediction, optimization, and design. We hope both the algorithms and the new RNA splicing design task will have a significant impact, serving the community on a variety of current tasks as well as a base for future developments.

## Methods

Our method, depicted in *Figure 1d*, involves two main components: a transformer-based splicing prediction model (TrASPr) and a BO algorithm (BOS) to design RNA with desired properties. We now turn to describe the two modeling components in order.

### TrASPr

### Pre-training RNA splice site BERT model

The base model for TrASPr is a six-layer BERT model pre-trained on human RNA splice sites (*Figure 1d*). Following the pre-training step, as in *Ji et al., 2021*, TrASPr takes an RNA sequence converted to 6-mer tokens as input, but instead of using the BERT default maximum length, we feed the model with 400 bases long sequences where the splice site (either 5' or 3' splice site, as shown in the illustration) is in the center.

For pre-training, we follow BERT (*Devlin et al., 2019*) in randomly choosing 15% of tokens but additionally mask the surrounding six tokens for each one to account for our overlapping 6-mer tokenization. We used standard masked autoencoder training, calculating the loss from the original 15% of tokens that were masked. The model is pre-trained for 110 k steps with a batch size of 40. The learning rate was set to 4e-4, and we used a linear scheduler with 10,000 warm-up steps.

### The TrASPr model and fine-tuning

The structure of TrASPr is depicted in *Figure 1d*. Each transformer encoder is a six-layer transformer with 12 heads. We set the hidden size of transformers as 768 for each layer. For any given AS event $e$, the input to TrASPr is a sequence composed of four sequences $S_e = \{S_e^i\}_{i=1}^4$ such that each $S_e^i$ covers the exonic and intronic regions surrounding one of the four splice sites involved in the exon skipping AS event $e$. Each $S_e^i$ is fed through a matching pre-trained transformer $T^i$, which also accepts additional event features $F_e = \{F_{e,i}\}$ (see below). The latent space representation from each transformer $T^i$, captured by their respective CLS tokens, is concatenated together along with the feature set $F_e$ and fed into two hidden layer MLPs with layer widths 3080 and 768.

### Event features

The additional feature set $F_e$ includes the exon and intron length information as discrete tokens (binned by length) and the tissue type. We also include discretized conservation generated based on the PhastCons score (*Siepel et al., 2005*) for each k-mer in the sequence. Exons generally have significantly higher conservation values, as these reflect selection pressure due to non-splicing related functions (coding for proteins). We, therefore, used a constant value for all exonic regions, which is the mean of all conservation scores across all intron positions in the training set, and kept the original values for the introns. In addition, we incorporated three types of indicators into the model: de novo, coding region, and frame-shift indicators. The de novo indicator specifies whether the middle exon is absent from existing annotations. The coding region indicators are five-digit binary flags to denote whether the middle exon falls within a coding region, a non-coding region, a start-coding region, an end-coding region, or a combination of multiple categories. The frame-shift indicator identifies whether the event results in a frame-shift AS outcome.

## Tissue representation with PCA

Tokens are widely used in language models to represent segments of sequence. However, token-based representations are inherently uninformative without context, and unseen tokens during training cannot be interpreted or utilized by the model. To address these issues, we propose an approach to learn the representation of tissues based on gene expression profiles. Specifically, we construct tissue representations using the expression levels of a curated list of 3344 RBPs. Each tissue vector was constructed with 15 samples from the GTEx dataset, with expression levels measured in TPMs (transcripts per million) obtained from the GTEx website. For our experiments, we selected eight tissues from GTEx (lung, heart, cerebellum, adrenal gland, liver, spleen, cortex, and EBV-transformed lymphocytes) and two cell lines from ENCODE (HepG2 and K562). The resulting representation vectors were then normalized using min–max normalization. Applying principal component analysis (PCA) (**Wold et al., 1987**) to these expression level vectors, we extract the top 50 components as the representation for the corresponding tissue. These PCA-derived representations are then incorporated into the feature vector by concatenating them to the sequence representation produced by the transformer encoders as shown in **Figure 1d**. The model is trained as described in the experiment section.

## Supervision

Since we are interested in learning splicing variations between different conditions, we define target variables that force the model to learn those (**Jha et al., 2017**). Specifically, based on the splicing outcome for an event $e \in E$ in two conditions $c, c' \in C$, where $E$ is all the cassette exon events and $T$ is all the tissue or cell line conditions, the target variables include:

$$\Psi_{e,c} = E[\Psi_{e,c}]$$

$$\Delta\Psi+_{e,c,c'} = |\max(\epsilon, E[\Delta\Psi_{e,c,c'}])|$$

$$\Delta\Psi-_{e,c,c'} = |\min(\epsilon, E[\Delta\Psi_{e,c,c'}])|$$

Here, $E[\Psi_{e,c}], E[\Delta\Psi_{e,c,c'}]$ represent the posterior expected values for PSI and Delta PSI (dPSI) as estimated by MAJIQ from the RNA-Seq experiments (**Vaquero-Garcia et al., 2016**). The $\Delta\Psi+_{e,c,c'}$ target captures events with increased inclusion level between tissue c and c' while $\Delta\Psi-_{e,c,c'}$ captures events with increased exclusion, incentivizing the model to focus its attention on splicing changes. To avoid the zero gradient issue, we use a random small number between 0.001 and 0.002 as $\epsilon$.

$$Loss = \sum_{e \in E,\, c,c' \in C} \left[ \mathcal{L}(\Psi_{e,c}, \hat{\Psi}_{e,c}) + \mathcal{L}(\Delta\Psi+_{e,c,c'}, \Delta\hat{\Psi}+_{e,c,c'}) + \mathcal{L}(\Delta\Psi-_{e,c,c'}, \Delta\hat{\Psi}-_{e,c,c'}) \right]$$

As shown above, we use the cross-entropy as the loss function $\mathcal{L}$, which performed better than the mean-squared error loss during our ablation studies. In the fine-tuning step, we train the model with a 2e-5 learning rate and batch size of 32 for ten epochs.

## Sequence design for splicing outcomes

Beyond supervised learning, we also demonstrate that TrASPr can be leveraged to solve sequence design problems. Given a genomic sequence context $S_e = (s_1, ..., s_n)$, made of a cassette exon $e$ and flanking intronic/exonic regions, TrASPr predicts for tissue $c$ the fraction of transcripts where exon $e$ is included or skipped over, $\Psi_c(S_e)$. This inclusion value can directly be used as the basis for optimization problems, where we seek to design new sequences $\tilde{S}_e$ that differ from $S_e$ only slightly but exhibit altered splicing outcomes. Formally, we define these optimization problems as:

$$\arg\min_{\tilde{S}_e} \Psi_c(\tilde{S}_e) \text{ s.t. } \text{lev}(\tilde{S}_e, S_e) \leq \tau$$

$$\text{or} \quad \arg\max_{\tilde{S}_e} \Psi_c(\tilde{S}_e) \text{ s.t. } \text{lev}(\tilde{S}_e, S_e) \leq \tau$$

(1)

Here, $\text{lev}(\tilde{S}_e, S_e)$ denotes the Levenshtein distance between $\tilde{S}_e$ and $S_e$. Solving the minimization problem is equivalent to finding a small perturbation (up to edit distance $\tau$) of $S_e$ that *reduces* inclusion in the target tissue $c$ by as much as possible. The maximization problem corresponds to *increasing* inclusion. In practice, we add additional constraints that $\forall c' \neq c$ and $\Psi_{c'}(\tilde{S}_e)$ cannot be reduced below

0.05. These additional constraints prevent an optimization routine from destroying splicing to such an extent that all inclusion levels are driven to zero.

To solve this optimization problem, we adapt recent work in LSBO for black-box optimization problems over structured and discrete inputs (*Maus et al., 2022*; *Stanton et al., 2022*; *Gligorijević et al., 2021*; *Moss et al., 2020*; *Winter et al., 2019*; *Sanchez-Lengeling and Aspuru-Guzik, 2018*; *Gómez-Bombarelli et al., 2018*; *Griffiths and Hernández-Lobato, 2020*; *Grosnit et al., 2021*). LSBO solves structured optimization problems using two primary components: (1) a deep VAE model and (2) a BO routine.

### Variational autoencoders for LSBO

In LSBO, we train a VAE that assists in reducing the discrete optimization problem over sequences $\mathcal{S}$ to a continuous optimization problem over the *latent space* of the VAE, $\mathcal{Z} \subset \mathbb{R}^d$. Leveraging the same data used to train TrASPr, we train a six-layer transformer encoder $\Phi : \mathcal{S} \to \mathcal{P}(\mathcal{Z})$ and a six-layer transformer *decoder* $\Gamma : \mathcal{Z} \to \mathcal{P}(\mathcal{S})$ (*Vaswani et al., 2017*). Here, $\mathcal{P}(\mathcal{Z})$ denotes the probability over the latent space $\mathcal{Z} \subset \mathbb{R}^d$, and $\mathcal{P}(\mathcal{S})$ denotes the probability over the sequence space $\mathcal{S}$. The encoder $\Phi(S_e)$ maps sequences $S_e$ onto a distribution over real-valued, 256-dimensional, continuous latent vectors $\mathbf{z} \in \mathbb{R}^{256}$. The decoder $\Gamma(\mathbf{z})$ reverses this process probabilistically. The parameters $\Phi$ and $\Gamma$ are trained so that we have $\Gamma(\Phi(S_e)) \approx S_e$. Because we only care about the output sequence $\tilde{S}_e$, here we abuse notation and denote the most probable sequence output from the decoder as $\Gamma(\mathbf{z})$. For optimization, the advantage the VAE provides is the ability to optimize over *latent vectors* $\mathbf{z} \in \mathbb{R}^{256}$ rather than directly over sequences $S_e$. This is because, for any $\mathbf{z}$ proposed by an optimization algorithm, we can evaluate $\Psi_c(\Gamma(\mathbf{z}))$. We therefore search for a $\tilde{\mathbf{z}}$ such that $\tilde{S}_e := \Gamma(\tilde{\mathbf{z}})$ is an optimal solution to the optimization problem.

### Bayesian optimization:

With the optimization problem in *Equation 1* reduced to a continuous problem over $\tilde{\mathbf{z}} \in \mathcal{Z}$, we can now apply standard continuous black-box optimization algorithms. BO (*Garnett, 2023*) is among the most well-studied of these approaches in the machine learning literature. In iteration $n$ of BO, we have a dataset $\mathcal{D}_n = \left\{ (\mathbf{z}_i, y_i) \right\}_{i=1}^{n}$ for which $y_i = \Psi_c(\Gamma(\mathbf{z}_i))$ is the known objective value. We train a surrogate model of the objective function using this data–most commonly a GP (*Rasmussen, 2003*)– and use this surrogate to inform a policy–commonly called an *acquisition function*–that determines what latent vectors $\mathbf{z}_{n+1}$ to consider next. Here, we used LOL-BO (*Maus et al., 2022*) as our base LS-BO algorithm. To accommodate the constraints in *Equation 1*, we modify LOL-BO to utilize SCBO (*Eriksson and Poloczek, 2021*) rather than TuRBO (*Eriksson et al., 2019*) as the underlying optimization routine. As with the objective, the Levenshtein constraint is evaluated on the decoded latent vectors: $\mathrm{lev}_{\mathcal{Z}}(\mathbf{z}, \mathbf{z}') = \mathrm{lev}(\Gamma(\mathbf{z}), \Gamma(\mathbf{z}'))$.

## Experimental validation

### Re-sequencing low-coverage cassette events with LSV-Seq

LSV-Seq was used to generate targeted libraries for this list of low-coverage cassette events prioritized for resequencing with TrASPr (*Yang et al., 2024*). Briefly, primers for LSV-Seq were designed using the Optimal Prime algorithm, which uses machine-learning models to optimize primer sequences for both specificity and yield. To perform LSV-Seq, the resulting primers were synthesized as a single combined pool of over 1000 primers and used in the first-strand reverse transcription reaction for RNA from each tissue of interest. Sequenceable libraries were created after additional reactions including second-strand synthesis, in vitro transcription, fragmentation, secondary reverse transcription, and final PCR amplification. The resulting libraries were aligned with STAR (*Dobin et al., 2013*). Quantification and visualization of psi values across tissues was performed with MAJIQ/VOILA (*Vaquero-Garcia et al., 2023*; *Vaquero-Garcia et al., 2016*).

### Targeting with dCas13d

In order to test regulatory elements predicted by TrASPr, we first trained it to predict differential splicing between the two cell lines: SH-SY5Y and HEK-293T. Selecting cassette exons with high confidence predictions for cell-line-specific splicing changes, we then selected regions for experimental testing using the following procedure. We randomly mutated sequences in the alternative exon and

flanking introns using a 6 bp sliding window. Each region in a window was mutated five times to avoid introducing new motifs, and the average predicted PSI was compared to the wild type. Finally, we selected top target regions based on predicted dPSI.

For experimental validation, predicted regulatory or negative control sequences were targeted with dCas13d. HEK-293T cells were obtained from ATCC (CRL-3216) and cultured in DMEM (Gibco, 10569010) supplemented with 10% (v/v) FBS (Thermo Fisher Scientific, A3160502) and 1 x penicillin-streptomycin (Thermo Fisher Scientific, 15140122). Cell lines were authenticated and tested negative for mycoplasma contamination by IDEXX BioAnalytics. For targeting, cells were co-transfected with vectors expressing dCas13d (pXR002, Addgene #109050) and guide RNA (cloned into a custom expression vector) using the CalPhos Mammalian Transfection Kit (Takara, 631312). Cells were collected 2 days later, and RNA was extracted using the Direct-zol RNA Purification Kit (Zymo Research, D2052). RNA was converted to cDNA with the LunaScript RT SuperMix (NEB, M3010L), and PCR for the splicing event was performed with the Q5 Hot Start High-Fidelity 2 X Master Mix (NEB, M0494S). PCR reactions were then visualized on a 2% agarose gel in 1 X lithium boric acid buffer (Faster Better Media, LB10-1) stained with SYBR Safe (Invitrogen, S33102). For all experiments, three biological replicates were performed.

Guide sequences were as follows:

- NT1 (non-targeting control 1): ATGATTATCCCGTACGCATGACATC
- NT2 (non-targeting control 2): TTACGATCCCCTCTATACGCGACCA
- ACIN1-nc1 (negative control): TTAGCATGAAAGGAGGTGTAACTAC
- ACIN1-g1: ATGGATCATCTCGTCCTCCGACTCA
- ACIN1-g2: AGGATGGATCATCTCGTCCTCCGAC
- ACIN1-g3: CTCAGGATGGATCATCTCGTCCTCC
- ACIN1-g4: TCCCTCAGGATGGATCATCTCGTCC
- ACIN1-g5: CACTCCCTCAGGATGGATCATCTCG
- PTK2-nc1 (negative control): GACATGAACACATCAACTGAGAACT
- PTK2-nc2 (negative control): GTGCGGGATATAGTTCAGATTGAGA
- PTK2-g1: GAATTTATCAGAGAAAGGGAAAAAA
- PTK2-g2: AAGGAATTTATCAGAGAAAGGGAAA
- PTK2-g3: TTAAAGGAATTTATCAGAGAAAGGG
- PTK2-g4: AAGGTGAGGAAAAGTTAAAGGAATT
- PTK2-g5: CGGAAGGTGAGGAAAAGTTAAAGGA

Primers for RT-PCR were as follows:

- ACIN1 forward: ACTTAGGCAGCGTCTGGAAC
- ACIN1 reverse: TTTCAGGATTGGTGCCTCCC
- PTK2 forward: CAGCTACAACGAGGGTGTCA
- PTK2 reverse: TGGGGCTGGCTGGATTTTAC

## Data
### Splice sites and alternative splicing events detection from GTEx
To pre-train the basic BERT RNA model, we first extract 1.5 million 400 bases long sequences around splice sites from the GENCODE v40 human pre-mRNA transcripts database. These splice junctions were extracted directly from the hg38 annotation regardless of whether those junctions are considered alternative or constitutive or their tissue specificity.

For tissue-specific splicing quantification, we used the GTEx dataset (*Aguet et al., 2020*) from which we selected six representative human tissues (heart-atrial appendage, cerebellum, lung, liver, spleen, and EBV-transformed lymphocytes). This RNA-Seq data was first aligned using STAR (*Dobin et al., 2013*) with default parameters. Next, we processed the resulting BAM files with MAJIQ to detect and quantify AS events. To construct a high-confidence set of cassette splicing events, we first built a splice graph for the six GTEx tissue groups using MAJIQ (*Vaquero-Garcia et al., 2023*), where we had 15 samples for each tissue group and filtered junctions existing in <20% samples. Then we ran MAJIQ dPSI and heterogeneity quantifier between 15 combinations of tissues to capture differential splicing patterns. From these outputs, VOILA modulizer was subsequently applied to classify cassette events into confident changing (dPSI ≥0.15 of heterogen quantified) and non-changing (dPSI <0.08 with confidence >0.7 of deltapsi quantified) categories. The resulting sets were merged and

further refined through a post-processing step that normalized PSI values, removed low-confidence or artifactual events, and retained only those supported across at least two tissues. Finally, sequence-based features were extracted with standardized exon region lengths, ensuring consistent input representations. To enrich the training data with negative examples, 'fake' skip junctions were added to constitutive exon events, and small random PSI values were assigned.

For AS events involving alternative 3' and 5' splice sites, we used the following procedure to generate a set of events to train and test on. For each alternative splice site type, we first scan for such alternative splice sites from the MAJIQ splice graphs for the analyzed tissues. If no qualified alternative splice site for the chosen tissues exists, other real alternative ones that are expressed in other tissues will be selected based on Ensembl annotation and MAJIQlopedia. These non-expressed alternative splice sites are assigned a small PSI value ~0.01. In the end, we got 1883 events for 5' and 1861 events for 3' splice site dataset.

## AS event filtering to avoid information leakage

When training and testing on AS events data, care must be taken to avoid testing on similar events that result in information leakage. This is especially important for large models that can easily memorize genomic sequences (*Schreiber et al., 2020*). One source of event similarity is paralog genes. Paralogs, derived from a common ancestral gene, may result in similar AS events exhibiting similar splicing patterns. To avoid this potential information leakage hazard, we first hide chromosomes (chr1, 3, 5, 7, 9 for experiments involved with Pangolin as in *Zeng and Li, 2022* and 8, 14 for the rest of experiments) for testing, then discarding test events similar to our training set. In addition, similar events in the training or test events can also occur due to overlapping cassette events detection. Specifically, cassette exons extracted from MAJIQ's splice graphs may overlap (e.g. different splice sites used to define the skipped exon). This may be useful for training on diverse exon/intron definitions, but care must be taken to avoid information leakage to the test data. To remove similar AS events due to either paralogs or overlapping AS event detection, we follow the procedure used in *Jha et al., 2017*. Specifically, exon sequence similarity was assessed using BLAT (*Kent, 2002*) with filters to identify the difference in length and the estimated similarity p-value. (BLAT settings maxLenDiff = 5, minPval = 0.0001 and minIdentity = 95). These settings account also for short exons with high similarity but enough divergence relative to their short length not to achieve a significant p-value.

## AS event quantification

Feature generation for 3' and 5' alternative splice site events followed processing steps similar to the ones used for cassette exon events, with the same filter parameters. Similar to the 'fake' skip junctions over constitutive exons, we also added here AS events where one of the two alternative splice-site junctions was not actually observed in the six GTEx tissues but was nonetheless annotated. For such junctions we first searched if they existed in the original data, then searched the Ensembl annotation D.B., then searched the MAJIQlopedia (*Quesnel-Vallières et al., 2024*) D.B. to see if they were detected across GTEx. Finally, if no such junction was found, we introduced a predicted 'fake' junction by selecting the best scoring position 400 up/downstream from the existing junction using SpliceAI.

We measure splicing across $c \in [1, \ldots, C]$ conditions for events $e \in [1, \ldots, E]$. Each AS event $e$ has a sequence $S_e$ comprised of four different regions, each centered around the respective splice site $S_e = \{S_e^1, S_e^2, S_e^3, S_e^4, \}$. Similarly, each event has a set of associated features, such as exon length, conservation, etc., denoted $F_e$. Splicing quantification for event $e$ in condition $c$ is denoted $\Psi_{e,c} \in [0,1]$ and differential splicing as $\Delta\Psi_{e,c,c'} \in [-1,1]$ accordingly. However, we frequently drop the event $e$ or condition $c$ index for brevity.

The above pipeline results in AS event quantification which can be directly fed as input for TrASPr. However, Pangolin and SpliceAI models do not define specific splicing events such as cassette exons. Instead, they use a 10 kb sequence window and predict if the center position is a 3' or 5' splice site. In addition, Pangolin predicts the 'splice usage' value for that genomic position, where the usage for a given genomic position in a given tissue is computed from RNA-Seq using SpliSER (*Dent et al., 2021*) during Pangolin's training. To make Pangolin and SpliceAI comparable for predicting AS event inclusion levels, we input the algorithms with 3' and 5' splice site of each alternative exon $e$. We tested different options to translate the splice sites scores by these algorithms to PSI and found the best

results were achieved by averaging the predictions of both 3' and 5' splice site usage. We also tried retraining Pangolin with actual PSI values from MAJIQ with similar results (see main text).

We used both the cassette exon and alternative splice site datasets to evaluate the ability of TrASPr to model tissue-specific splicing. To ensure robust assessment, we removed paralogous and overlapping events through chromosome masking and sequence similarity filtering, resulting in independent datasets for training TrASPr and benchmarking against existing models. AS event data is collected in alternative_3.tsv and alternative_5.tsv.

## Splice site swap and disruption for RNA sequence design

The splice site plays a critical role in AS. Identifying and then restoring them represents a crucial milestone for TrASPr and BOS. To assess how TrASPr responds to modifications in splice site strength, we scored all donor and acceptor splice sites using MaxEnt (*Yeo and Burge, 2003*). Based on these scores, we classified splice sites as strong or weak by selecting the top 15% (5' score > 10.17, 3' score > 10.83) and bottom 15% (5' score < 5.95, 3' score < 5.52) from the distribution of splice site scores in the test set. For low inclusion sequences (PSI < 0.15), we substituted their 3' and 5' splice sites in the middle exon with strong splice sites. Similarly, for high-inclusivity sequences (PSI > 0.85), we replaced the existing splice sites with weaker ones. Our goal was to observe TrASPr's predicted changes in PSI before and after each replacement.

After confirming TrASPr's ability on weak and strong splice site recognition, we further designed the experiment to test if BOS is able to recover the disrupted splice sites. In this case, we followed the same strategy to select high-inclusive sequences and strong splice sites. These strong 3' or 5' splice sites were disrupted through random mutations. Then, we randomly selected eight sequences each for the 3' and 5' splice sites as starters for generative models aimed at increasing PSI. In the end, the successfully generated sequences will be scored by MaxEnt to determine if their PSI is increased by generating strong splice sites.

### Mutations and knockdown data

To evaluate the capability of TrASPr and BOS to predict or suggest mutations, we curated four other datasets. The first dataset is the ENCODE database for RBP KD in human cell lines (*Van Nostrand et al., 2020*), where we focused on three well-studied RBPs (TIA1, PTBP1, QKI). This data resulted in a list of 59 putative RBP regulatory targets for which we could 'remove' the effect of these RBPs on the set of their AS targets by randomly mutating their identified binding motifs. This set was then supplemented with nine validated targets of RBFOX from a recent study (*Maurin et al., 2023*). In the prediction results, 'change' and 'no change' are determined based on a threshold set at the 95th percentile of observed effects from random mutations. To mitigate the positional effect of mutations, we utilized the same relative distance of the RBP binding sites from splice sites in our original set. For each position, we randomly select 100 events from the test set and performed five different random mutations. To ensure a fair comparison across different models, we evaluated all models on the same random mutation dataset and calculated their 95th percentile thresholds, which correspond to the following values. TrASPr: $\Delta PSI>0.019$, SpliceAI: $\Delta Usage>0.043$, Pangolin: $\Delta Usage>0.047$. We note that these threshold values are specific to the exons and RBP sites we tested, aimed to create a uniform testing procedure for all methods. Finally, we included two additional datasets to assess splicing outcome predictions in the presence of genetic mutations. One is a recent high-throughput assay with 6106 mutation combinations around exon 2 from the CD19 gene (*Cortés-López et al., 2022*). This data is typical of assays that measure many mutations in a clinically relevant event, not necessarily tissue-specific ones. For capturing a tissue-specific event, we also included low-throughput experiments from a mini-gene reporter assay where the effect of mutating several regions upstream of the neuronal-specific exon 16 of the mouse Daam1 gene was tested (*Barash et al., 2010*).

## Acknowledgements

We thank members of the Barash lab for helpful comments and discussions. This work was supported by NIH grants R01-LM013437, GM-147739, NSF AirFoundry DBI-2400135, CureBRCA and the Basser Center for BRCA pilot grant to YB; NIGMS DP2GM146251 to PC; NSF IIS-2145644 to JRG; NIGMS T32GM156697 to BDW-M.

## Additional information

### Funding

| Funder | Grant reference number | Author |
|---|---|---|
| National Library of Medicine | LM013437 | Yoseph Barash |
| National Institute of General Medical Sciences | GM-147739 | Yoseph Barash |
| CureBRCA | | Yoseph Barash |
| Basser Center for BRCA | | Yoseph Barash |
| National Science Foundation | DBI-2400135 | Yoseph Barash |
| National Science Board | IIS-2145644 | Jake R Gardner |
| National Institute of General Medical Sciences | T32GM156697 | Benjamin D Wales-McGrath |
| National Institute of General Medical Sciences | DP2GM146251 | Peter Choi |

The funders had no role in study design, data collection and interpretation, or the decision to submit the work for publication.

### Author contributions

Di Wu, Data curation, Software, Formal analysis, Investigation, Methodology, Writing – original draft; Natalie Maus, Software, Methodology, Writing – original draft; Anupama Jha, Conceptualization, Data curation, Methodology, Writing – review and editing; Kevin Yang, Anna Tangiyan, Investigation; Benjamin D Wales-McGrath, Formal analysis, Investigation, Writing – original draft; San Jewell, Data curation, Software; Peter Choi, Supervision, Investigation; Jake R Gardner, Conceptualization, Resources, Supervision, Methodology, Writing – review and editing; Yoseph Barash, Conceptualization, Resources, Formal analysis, Supervision, Funding acquisition, Methodology, Writing – original draft, Writing – review and editing

### Author ORCIDs

Di Wu ⓘ https://orcid.org/0000-0003-2002-2883
Yoseph Barash ⓘ https://orcid.org/0000-0003-3005-5048

Reviewer #1 (Public review): https://doi.org/10.7554/eLife.106043.3.sa1
Reviewer #2 (Public review): https://doi.org/10.7554/eLife.106043.3.sa2
Author response https://doi.org/10.7554/eLife.106043.3.sa3

## Additional files

### Supplementary files

MDAR checklist

### Data availability

Source code and data is available on https://bitbucket.org/biociphers/traspr/src/main/ (copy archived at *biociphers, 2026*). It contains sample data, source code of the model, and an executable notebook version of the paper to reproduce all paper figures. Pre-train and fine-tuning model weights can be found on https://huggingface.co/MagicSign/TrASPr under a license for non-commercial use or educational purposes specified in the repository. The splicing quantification from the GTEx dataset used for fine-tuning the model can be downloaded from https://huggingface.co/datasets/MagicSign/TrASPr_GTEx_data.

The following datasets were generated:

| Author(s) | Year | Dataset title | Dataset URL | Database and Identifier |
|---|---|---|---|---|
| Wu D | 2025 | TrASPr | https://huggingface.co/MagicSign/TrASPr | HuggingFace, TrASPr |
| Wu D | 2025 | TrASPr_GTEx_data | https://huggingface.co/datasets/MagicSign/TrASPr_GTEx_data | HuggingFace, TrASPr_GTEx_data |

The following previously published datasets were used:

| Author(s) | Year | Dataset title | Dataset URL | Database and Identifier |
|---|---|---|---|---|
| The GTEx Consortium | 2020 | Genotype-Tissue Expression (GTEx) project, version 8 | https://www.ncbi.nlm.nih.gov/projects/gap/cgi-bin/study.cgi?study_id=phs000424.v8.p2 | dbGAP, phs000424.v8.p2 |
| Cortés-López M, Schulz L, Enculescu M, Paret C, Spiekermann B, Quesnel-Vallières M, Torres-Diz M, Unic S, Busch A, Orekhova A, Kuban M, Mesitov M, Mulorz MM, Shraim R, Kielisch F, Faber J, Barash Y, Thomas-Tikhonenko A, Zarnack K, Legewie S, König J | 2022 | Data from: Mutations and RNA-binding proteins controlling CD19 splicing and CART-19 therapy resistance | https://www.ncbi.nlm.nih.gov/geo/query/acc.cgi?acc=GSE182894 | NCBI Gene Expression Omnibus, GSE182894 |
| Maurin M, Ranjouri M, Megino-Luque C, Newberg JY, Du D, Martin K, Miner RE, Prater MS, Centeno B, Pruett-Miller SM, Stewart P, Fleming JB, Yu X, Bravo-Cordero JJ, Guccione E, Black MA, Mann KM, Wee DKB | 2023 | Data from: Expression data from human pancreatic cancer cell lines and mouse orthotopic tumors generated using human PDAC cell lines replete and depleted for RBFOX2 | https://www.ncbi.nlm.nih.gov/geo/query/acc.cgi?acc=GSE211435 | NCBI Gene Expression Omnibus, GSE211435 |
| Van Nostrand EL, Freese P, Pratt GA, Wang X, Wei X, Xiao R, Blue SM, Chen J-Y, Cody NAL, Dominguez D, Olson S, Sundararaman B, Zhan L, Bazile C, Bouvrette LPB, Bergalet J, Duff MO, Garcia KE, Gelboin-Burkhart C, Hochman M, Lambert NJ, Li H, McGurk MP, Nguyen TB, Palden T, Rabano I, Sathe S, Stanton R, Su A, Wang R, Yee BA, Zhou B, Louie AL, Aigner S, X-D Fu, Lécuyer E, Burge CB, Graveley BR, Yeo GW | 2020 | A large-scale binding and functional map of human RNA-binding proteins | https://www.encodeproject.org/encore-matrix/?type=Experiment&status=released&internal_tags=ENCORE | ENCODE, ENCSR057GCF |

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
