## [Editor Report · eLife Assessment]

TrASPr is an **important** contribution that leverages transformer models focused on regulatory regions to enhance predictions of tissue-specific splicing events. The revisions strengthen the manuscript by clarifying methodology and expanding analyses across exon and intron sizes, and the evidence supporting TrASPr's predictive performance is **compelling**. This work will be of interest to researchers in computational genomics and RNA biology, offering an improved model for splicing prediction and a promising approach to RNA sequence design.

---

## [Referee Report · Reviewer #1 (Public review)]

Summary

The authors propose a transformer-based model for prediction of condition- or tissue-specific alternative splicing and demonstrate its utility in design of RNAs with desired splicing outcomes, which is a novel application. The model is compared to relevant exising approaches (Pangolin and SpliceAI) and the authors clearly demonstrate its advantage. Overall, a compelling method that is well thought out and evaluated.

Strengths:

(1) The model is well thought out: rather than modeling a cassette exon using a single generic deep learning model as has been done e.g. in SpliceAI and related work, the authors propose a modular architecture that focuses on different regions around a potential exon skipping event, which enables the model to learn representations that are specific to those regions. Because each component in the model focuses on a fixed length short sequence segment, the model can learn position-specific features. Furthermore, the architecture of the model is designed to model alternative splicing events, whereas Pangolin and SpliceAI are focused on modeling individual splice junctions, which is an easier problem.

(2) The model is evaluated in a rigorous way - it is compared to the most relevant state-of-the-art models, uses machine learning best practices, and an ablation study demonstrates the contribution of each component of the architecture.

(3) Experimental work supports the computational predictions: Regulatory elements predicted by the model were experimentally verified; novel tissue-specific cassette exons were verified by LSV-seq.

(4) The authors use their model for sequence design to optimize splicing outcome, which is a novel application.

Weaknesses:

None noted.

---

## [Referee Report · Reviewer #2 (Public review)]

Summary:

The authors present a transformer-based model, TrASPr, for the task of tissue-specific splicing prediction (with experiments primarily focused on the case of cassette exon inclusion) as well as an optimization framework (BOS) for the task of designing RNA sequences for desired splicing outcomes.

For the first task, the main methodological contribution is to train four transformer-based models on the 400bp regions surrounding each splice site, the rationale being that this is where most splicing regulatory information is. In contrast, previous work trained one model on a long genomic region. This new design should help the model capture more easily interactions between splice sites. It should also help in cases of very long introns, which are relatively common in the human genome.

TrASPr's performance is evaluated in comparison to previous models (SpliceAI, Pangolin, and SpliceTransformer) on numerous tasks including splicing predictions on GTEx tissues, ENCODE cell lines, RBP KD data, and mutagenesis data. The scope of these evaluations is ambitious; however, significant details on most of the analyses are missing, making it difficult to evaluate the strength of evidence.

In the second task, the authors combine Latent Space Bayesian Optimization (LSBO) with a Transformer-based variational auto encoder to optimize RNA sequences for a given splicing-related objective function. This method (BOS) appears to be a novel application of LSBO, with promising results on several computational evaluations and the potential to be impactful on sequence design for both splicing-related objectives and other tasks. However, comparison of BOS against existing methods for sequence design is lacking.

Strengths:

- A novel machine learning model for an important problem in RNA biology with excellent prediction accuracy.

- Instead of being based on a generic design as in previous work, the proposed model incorporates biological domain knowledge (that regulatory information is concentrated around splice sites). This way of using inductive bias can be important to future work on other sequence-based prediction tasks.

Weaknesses:

- Most of the analyses presented in the manuscript are described in broad strokes and are often confusing. As a result, it is difficult to assess the significance of the contribution.

- As more and more models are being proposed for splicing prediction (SpliceAI, Pangolin, SpliceTransformer, TrASPr), there is a need for establishing standard benchmarks, similar to those in computer vision (ImageNet). Without such benchmarks, it is exceedingly difficult to compare models.

*This point is now addressed in the revision *

*Moreover, datasets have been made available by the authors on BitBucket. *

- Related to the previous point, as discussed in the manuscript, SpliceAI and Pangolin are not designed to predict PSI of cassette exons. Instead, they assign a "splice site probability" to each nucleotide. Converting this to a PSI prediction is not obvious, and the method chosen by the authors (averaging the two probabilities (?)) is likely not optimal. It would interesting to see what happens if an MLP is used on top of the four predictions (or the outputs of the top layers) from SpliceAI/Pangolin. This could also indicate where the improvement in TrASPr comes from: is it because TrASPr combines information from all four splice sites? Also consider fine-tuning Pangolin on cassette exons only (as you do for your model).

*This point is still not addressed in the revision. *

- L141, "TrASPr can handle cassette exons spanning a wide range of window sizes from 181 to 329,227 bases-thanks to its multi-transformer architecture." This is reported to be one of the primary advantages compared to existing models. Additional analysis should be included on how TrASPr performs across varying exon and intron sizes, with comparison to SpliceAI, etc.

Added after revision: The authors have added additional analyses of performance based on both the length of the exon under consideration and the total length of the surrounding intronic contexts. The result that TrASPr performs well across various context sizes (i.e., the length of the sequence between the upstream and downstream exons, ranging from <1k to >10k) is highly encouraging and supports the claim that most of the sequence-based splicing logic is located proximal to the splice sites. It is also noteworthy that TrASPr performs well for exons longer than 200, suggesting that most of the "regulatory code" is present at the exon boundaries rather than in its center (which TrASPr is blind to).

Additionally, Pearson correlation is used as the sole performance metric in many analyses (e.g., Fig 2 - Supp 2). The authors should consider alternative accuracy metrics, such as RMSE, which better convey the magnitude of prediction error and are more easily comparable across datasets. Pearson correlation may also be more sensitive to outliers on the smaller samples that arise when binning sequences.

- L171, "training it on cassette exons". This seems like an important point: previous models were trained mostly on constitutive exons, whereas here the model is trained specifically on cassette exons. This should be discussed in more detail.

* Our initial comment was incorrect, as pointed out by the authors. *

- L214, ablations of individual features are missing.

* This was addressed in the revision. *

- L230, "ENCODE cell lines", it is not clear why other tissues from GTEx were not included

* This was addressed in the revision. *

- L239, it is surprising that SpliceAI performs so badly, and might suggest a mistake in the analysis. Additional analysis and possible explanations should be provided to support these claims. Similarly for the complete failure of SpliceAI and Pangolin shown in Fig 4d.

* The authors should consider adding SpliceAI/Pangolin predictions for the alternative 5' and 3' splice site selection tasks (and code for related analyses) to the BitBucket repository.*

- BOS seems like a separate contribution that belongs in a separate publication. Instead, consider providing more details on TrASPr.

*Minor comment added after revision: regarding the author response that "A completely independent evaluation would have required a high-throughput experimental system to assess designs, which is beyond the scope of the current paper.":

It's not clear why BOS cannot be evaluated as a separate contribution by instead using different "teacher" models instead of TrASPr. Additionally, BOS lacks evaluation against existing methods for sequence optimization. *

- The authors should consider evaluating BOS using Pangolin or SpliceTransformer as the oracle, in order to measure the contribution to the sequence generation task provided by BOS vs TrASPr.

* See comment above *

---

## [Author Response]

The following is the authors’ response to the original reviews

A point by point response included below. Before we turn to that we want to note one change that we decided to introduce, related to generalization on unseen tissues/cell types (Figure 3a in the original submission and related question by Reviewer #2 below). This analysis was based on adding a latent “RBP state” representation during learning of condition/tissue specific splicing. The “RBP state” per condition is captured by a dedicated encoder. Our original plan was to have a paper describing a new RBP-AE model we developed in parallel, which also served as the base to capture this “RBP State”. However, we got delayed in getting this second paper finalized (it was led by other lab members, some of whom have already left the lab). This delay affected the TrASPr manuscript as TrASPr’s code should be available and analysis reproducible upon publication. After much deliberation, we decided that in order to comply with reproducibility standards while not self scooping the RBP-AE paper, we eventually decided to take out the RBP-AE and replace it with a vanilla PCA based embedding for the “RBP-State”. The PCA approach is simpler and reproducible, based on linear transformation of the RBPs expression vector into a lower dimension. The qualitative results included in Figure 3a still hold, and we also produced the new results suggested by Reviewer #2 in other GTEX tissues with this PCA based embedding (below).

We don’t believe the switch to PCA based embedding should have any bearing on the current manuscript evaluation but wanted to take this opportunity to explain the reasoning behind this additional change.

**Public Reviews:**

**Reviewer #1 (Public review):**
Summary:The authors propose a transformer-based model for the prediction of condition - or tissue-specific alternative splicing and demonstrate its utility in the design of RNAs with desired splicing outcomes, which is a novel application. The model is compared to relevant existing approaches (Pangolin and SpliceAI) and the authors clearly demonstrate its advantage. Overall, a compelling method that is well thought out and evaluated.Strengths:(1) The model is well thought out: rather than modeling a cassette exon using a single generic deep learning model as has been done e.g. in SpliceAI and related work, the authors propose a modular architecture that focuses on different regions around a potential exon skipping event, which enables the model to learn representations that are specific to those regions. Because each component in the model focuses on a fixed length short sequence segment, the model can learn position-specific features. Another difference compared to Pangolin and SpliceAI which are focused on modeling individual splice junctions is the focus on modeling a complete alternative splicing event.(2) The model is evaluated in a rigorous way - it is compared to the most relevant state-of-the-art models, uses machine learning best practices, and an ablation study demonstrates the contribution of each component of the architecture.(3) Experimental work supports the computational predictions.(4) The authors use their model for sequence design to optimize splicing outcomes, which is a novel application.

We wholeheartedly thank Reviewer #1 for these positive comments regarding the modeling approach we took to this task and the evaluations we performed. We have put a lot of work and thought into this and it is gratifying to see the results of that work acknowledged like this.

Weaknesses:No weaknesses were identified by this reviewer, but I have the following comments:(1) I would be curious to see evidence that the model is learning position-specific representations.

This is an excellent suggestion to further assess what the model is learning. To get a better sense of the position-specific representation we performed the following analyses:

(1) Switching the transformers relative order: All transformers are pretrained on 3’ and 5’ splice site regions before fine-tunning for the PSI and dPSI prediction task. We hypothesized that if relative position is important, switching the order of the transformers would make a large difference on prediction accuracy. Indeed if we switch the 3’ and 5’ we see as expected a severe drop in performance, with Pearson correlation on test data dropping from 0.82 to 0.11. Next, we switched the two 5’ and 3’ transformers, observing a drop to 0.65 and 0.78 respectively. When focusing only on changing events the drop was from 0.66 to 0.54 (for 3’ SS transformers), 0.48 (for 5’ SS transformers), and 0.13 (when the 3’ and 5’ transformers flanking the alternative exon were switched).

(2) Position specific effect of RBPs: We wanted to test whether the model is able to learn position specific effects for RBPs. For this we focused on two RBPs, FOX (a family of three highly related RBPs), and QKI, both have a relatively well defined motif, known condition and position specific effect identified via RBP KD experiments combined with CLIP experiments (e.g. PMID: 23525800, PMID: 24637117, PMID: 32728246). For each, we randomly selected 40 highly and 40 lowly included cassette exons sequences. We then ran in-silico mutagenesis experiments where we replaced small windows of sequences with the RBP motifs (80 for RBFOX and 80 for QKI), then compared TrASPR’s predictions for the average predictions for 5 random sequences inserted in the same location. The results of this are now shown in Figure 4 Supp 3, where the y-axis represents the dPSI effect per position (x-axis), and the color represents the percentile of observed effects over inserting motifs in that position across all 80 sequences tested. We see that both RBPs have strong positional preferences for exerting a strong effect on the alternative exon. We also see differences between binding upstream and downstream of the alternative exon. These results, learned by the model from natural tissue-specific variations, recapitulate nicely the results derived from high-throughput experimental assays. However, we also note that effects were highly sequence specific. For example, RBFOX is generally expected to increase inclusion when binding downstream of the alternative exon and decrease inclusion when binding upstream. While we do observe such a trend we also see cases where the opposite effects are observed. These sequence specific effects have been reported in the literature but may also represent cases where the model errs in the effect’s direction. We discuss these new results in the revised text.

(3) Assessing BOS sequence edits to achieve tissue-specific splicing: Here we decided to test whether BOS edits in intronic regions (at least 8b away from the nearest splice site) are important for the tissue-specific effect. The results are now included in Figure 6 Supp 1, clearly demonstrating that most of the neuronal specific changes achieved by BOS were based on changing the introns, with a strong effect observed for both up and downstream intron edits.

(2) The transformer encoders in TrASPr model sequences with a rather limited sequence size of 200 bp; therefore, for long introns, the model will not have good coverage of the intronic sequence. This is not expected to be an issue for exons.

The reviewer is raising a good question here. On one hand, one may hypothesize that, as the reviewer seems to suggest, TrASPr may not do well on long introns as it lacks the full intronic sequence.

Conversely, one may also hypothesize that for long introns, where the flanking exons are outside the window of SpliceAI/Pangolin, TrASPr may have an advantage.

Given this good question and a related one by Reviewer #2, we divided prediction accuracy by intron length and the alternative exon length.

For short exons (<100bp) we find TrASPr and Pangolin perform similarly, but for longer exons, especially those > 200, TrASPr results are better. When dividing samples by the total length of the upstream and downstream intron, we find TrASPr outperform all other models for introns of combined length up to 6K, but Pangolin gets better results when the combined intron length is over 10K. This latter result is interesting as it means that contrary to the second hypothesis laid out above, Pangolin’s performance did not degrade for events where the flanking exons were outside its field of view. We note that all of the above holds whether we assess all events or just cases of tissue specific changes. It is interesting to think about the mechanistic causes for this. For example, it is possible that cassette exons involving very long introns evoke a different splicing mechanism where the flanking exons are not as critical and/or there is more signal in the introns which is missed by TrASPr. We include these new results now as Figure 2 - Supp 1,2 and discuss these in the main text.

(3) In the context of sequence design, creating a desired tissue- or condition-specific effect would likely require disrupting or creating motifs for splicing regulatory proteins. In your experiments for neuronal-specific Daam1 exon 16, have you seen evidence for that? Most of the edits are close to splice junctions, but a few are further away.

That is another good question. Regarding Daam1 exon 16, in the original paper describing the mutation locations some motif similarities were noted to PTB (CU) and CUG/Mbnl-like elements (Barash et al Nature 2010). In order to explore this question beyond this specific case we assessed the importance of intronic edits by BOS to achieve a tissue specific splicing profile - see above.

(4) For sequence design, of tissue- or condition-specific effect in neuronal-specific Daam1 exon 16 the upstream exonic splice junction had the most sequence edits. Is that a general observation? How about the relative importance of the four transformer regions in TrASPr prediction performance?

This is another excellent question. Please see new experiments described above for RBP positional effect and BOS edits in intronic regions which attempt to give at least partial answers to these questions. We believe a much more systematic analysis can be done to explore these questions but such evaluation is beyond the scope of this work.

(5) The idea of lightweight transformer models is compelling, and is widely applicable. It has been used elsewhere. One paper that came to mind in the protein realm:Singh, Rohit, et al. "Learning the language of antibody hypervariability." Proceedings of the National Academy of Sciences 122.1 (2025): e2418918121.

We definitely do not make any claim this approach of using lighter, dedicated models instead of a large ‘foundation’ model has not been taken before. We believe Rohit et al mentioned above represents a somewhat different approach, where their model (AbMAP) fine-tunes large general protein foundational models (PLM) for antibody-sequence inputs by supervising on antibody structure and binding specificity examples. We added a description of this modeling approach citing the above work and another one which specifically handles RNA splicing (intron retention, PMID: 39792954).

**Reviewer #2 (Public review):**
Summary:The authors present a transformer-based model, TrASPr, for the task of tissue-specific splicing prediction (with experiments primarily focused on the case of cassette exon inclusion) as well as an optimization framework (BOS) for the task of designing RNA sequences for desired splicing outcomes.For the first task, the main methodological contribution is to train four transformer-based models on the 400bp regions surrounding each splice site, the rationale being that this is where most splicing regulatory information is. In contrast, previous work trained one model on a long genomic region. This new design should help the model capture more easily interactions between splice sites. It should also help in cases of very long introns, which are relatively common in the human genome.TrASPr's performance is evaluated in comparison to previous models (SpliceAI, Pangolin, and SpliceTransformer) on numerous tasks including splicing predictions on GTEx tissues, ENCODE cell lines, RBP KD data, and mutagenesis data. The scope of these evaluations is ambitious; however, significant details on most of the analyses are missing, making it difficult to evaluate the strength of the evidence. Additionally, state-of-the-art models (SpliceAI and Pangolin) are reported to perform extremely poorly in some tasks, which is surprising in light of previous reports of their overall good prediction accuracy; the reasoning for this lack of performance compared to TrASPr is not explored.In the second task, the authors combine Latent Space Bayesian Optimization (LSBO) with a Transformer-based variational autoencoder to optimize RNA sequences for a given splicing-related objective function. This method (BOS) appears to be a novel application of LSBO, with promising results on several computational evaluations and the potential to be impactful on sequence design for both splicing-related objectives and other tasks.

We thank Reviewer #2 for this detailed summary and positive view of our work. It seems the main issue raised in this summary regards the evaluations: The reviewer finds details of the evaluations missing and the fact that SpliceAI and Pangolin perform poorly on some of the tasks to be surprising. We made a concise effort to include the required details, including code and data tables. In short, some of the concerns were addressed by adding additional evaluations, some by clarifying missing details, and some by better explaining where Pangolin and SpliceAI may excel vs. settings where these may not do as well. More details are given below.

Strengths:(1) A novel machine learning model for an important problem in RNA biology with excellent prediction accuracy.(2) Instead of being based on a generic design as in previous work, the proposed model incorporates biological domain knowledge (that regulatory information is concentrated around splice sites). This way of using inductive bias can be important to future work on other sequence-based prediction tasks.Weaknesses:(1) Most of the analyses presented in the manuscript are described in broad strokes and are often confusing. As a result, it is difficult to assess the significance of the contribution.

We made an effort to make the tasks be specific and detailed, including making the code and data of those available. We believe this helped improve clarity in the revised version.

(2) As more and more models are being proposed for splicing prediction (SpliceAI, Pangolin, SpliceTransformer, TrASPr), there is a need for establishing standard benchmarks, similar to those in computer vision (ImageNet). Without such benchmarks, it is exceedingly difficult to compare models. For instance, Pangolin was apparently trained on a different dataset (Cardoso-Moreira et al. 2019), and using a different processing pipeline (based on SpliSER) than the ones used in this submission. As a result, the inferior performance of Pangolin reported here could potentially be due to subtle distribution shifts. The authors should add a discussion of the differences in the training set, and whether they affect your comparisons (e.g., in Figure 2). They should also consider adding a table summarizing the various datasets used in their previous work for training and testing. Publishing their training and testing datasets in an easy-to-use format would be a fantastic contribution to the community, establishing a common benchmark to be used by others.

There are several good points to unpack here. Starting from the last one, we very much agree that a standard benchmark will be useful to include. For tissue specific splicing quantification we used the GTEx dataset from which we select six representative human tissues (heart, cerebellum, lung, liver, spleen, and EBV-transformed lymphocytes). In total, we collected 38394 cassette exon events quantified across 15 samples (here a ‘sample’ is a cassette exon quantified in two tissues) from the GTEx dataset with high-confidence quantification for their PSIs based on MAJIQ. A detailed description of how this data was derived is now included in the Methods section, and the data itself is made available via the bitbucket repository with the code.

Next, regarding the usage of different data and distribution shifts for Pangolin: The reviewer is right to note there are many differences between how Pangolin and TrASPr were trained. This makes it hard to determine whether the improvements we saw are not just a result of different training data/labels. To address this issue, we first tried to finetune the pre-trained Pangolin with MAJIQ’s PSI dataset: we use the subset of the GTEx dataset described above, focusing on the three tissues analyzed in Pangolin’s paper—heart, cerebellum, and liver—for a fair comparison. In total, we obtained 17,218 events, and we followed the same training and test split as reported in the Pangolin paper. We got Pearson: 0.78 Spearman: 0.68 which are values similar to what we got without this extra fine tuning. Next, we retrained Pangolin from scratch, with the full tissues and training set used for TrASPr, which was derived from MAJIQ’s quantifications. Since our model only trained on human data with 6 tissues at the same time, we modified Pangolin from original 4 splice site usage outputs to 6 PSI outputs. We tried to take the sequence centered with the first or the second splice site of the mid exon. This test resulted in low performance (3’ SS: pearson 0.21 5’ SS: 0.26.).

The above tests are obviously not exhaustive but their results suggest that the differences we observe are unlikely to be driven by distribution shifts. Notably, the original Pangolin was trained on much more data (four species, four tissues each, and sliding windows across the entire genome). This training seems to be important for performance while the fact we switched from Pangolin’s splice site usage to MAJIQ’s PSI was not a major contributor. Other potential reasons for the improvements we observed include the architecture, target function, and side information (see below) but a complete delineation of those is beyond the scope of this work.

(3) Related to the previous point, as discussed in the manuscript, SpliceAI, and Pangolin are not designed to predict PSI of cassette exons. Instead, they assign a "splice site probability" to each nucleotide. Converting this to a PSI prediction is not obvious, and the method chosen by the authors (averaging the two probabilities (?)) is likely not optimal. It would be interesting to see what happens if an MLP is used on top of the four predictions (or the outputs of the top layers) from SpliceAI/Pangolin. This could also indicate where the improvement in TrASPr comes from: is it because TrASPr combines information from all four splice sites? Also, consider fine-tuning Pangolin on cassette exons only (as you do for your model).

Please see the above response. We did not investigate more sophisticated models that adjust Pangolin’s architecture further as such modifications constitute new models which are beyond the scope of this work.

(4) L141, "TrASPr can handle cassette exons spanning a wide range of window sizes from 181 to 329,227 bases - thanks to its multi-transformer architecture." This is reported to be one of the primary advantages compared to existing models. Additional analysis should be included on how TrASPr performs across varying exon and intron sizes, with comparison to SpliceAI, etc.

This was a good suggestion, related to another comment made by Reviewer #1. Please see above our response to them with a breakdown by exon/intron length.

(5) L171, "training it on cassette exons". This seems like an important point: previous models were trained mostly on constitutive exons, whereas here the model is trained specifically on cassette exons. This should be discussed in more detail.

Previous models were not trained exclusively on constitutive exons and Pangolin specifically was trained with their version of junction usage across tissues. That said, the reviewer’s point is valid (and similar to ones made above) about a need to have a matched training/testing and potential distribution shifts. Please see response and evaluations described above.

(6) L214, ablations of individual features are missing.

These were now added to the table which we moved to the main text (see table also below).

(7) L230, "ENCODE cell lines", it is not clear why other tissues from GTEx were not included.

Good question. The task here was to assess predictions in unseen conditions, hence we opted to test on completely different data of human cell lines rather than additional tissue samples. Following the reviewers suggestion we also evaluated predictions on two additional GTEx tissues, Cortex and Adrenal Gland. These new results, as well as the previous ones for ENCODE, were updated to use the PCA based embedding of “RBP-State” as described above. We also compared the predictions using the PCA based embedding of the “RBP-State” to training directly on data (not the test data of course) from these tissues. See updated Figure 3a,b. Figure 3 Supp 1,2.

(8) L239, it is surprising that SpliceAI performs so badly, and might suggest a mistake in the analysis. Additional analysis and possible explanations should be provided to support these claims. Similarly, the complete failure of SpliceAI and Pangolin is shown in Figure 4d.

Line 239 refers to predicting relative inclusion levels between competing 3’ and 5’ splice sites. We admit we too expected this to be better for SpliceAI and Pangolin but we were not able to find bugs in our analysis (which is all made available for readers and reviewers alike). Regarding this expectation to perform better, first we note that we are not aware of a similar assessment being done for either of those algorithms (i.e. relative inclusion for 3’ and 5’ alternative splice site events). Instead, our initial expectation, and likely the reviewer’s as well, was based on their detection of splice site strengthening/weakening due to mutations, including cryptic splice site activation. More generally though, it is worth noting in this context that given how SpliceAI, Pangolin and other algorithms have been presented in papers/media/scientific discussions, we believe there is a potential misperception regarding tasks that SpliceAI and Pangolin excel at vs other tasks where they should not necessarily be expected to excel. Both algorithms focus on cryptic splice site creation/disruption. This has been the focus of those papers and subsequent applications. While Pangolin added tissue specificity to SpliceAI training, the authors themselves admit “...predicting differential splicing across tissues from sequence alone is possible but remains a considerable challenge and requires further investigation”. The actual performance on this task is not included in Pangolin’s main text, but we refer Reviewer #2 to supplementary figure S4 in the Pangolin manuscript to get a sense of Pangolin’s reported performance on this task. Similar to that, Figure 4d in our manuscript is for predicting ‘tissue specific’ regulators. We do not think it is surprising that SpliceAI (tissue agnostic) and Pangolin (slight improvement compared to SpliceAI in tissue specific predictions) do not perform well on this task. Similarly, we do not find the results in Figure 4C surprising either. These are for mutations that slightly alter inclusion level of an exon, not something SpliceAI was trained on - SpiceAI was trained on genomic splice sites with yes/no labels across the genome. As noted elsewhere in our response, re-training Pangolin on this mutagenesis dataset results in performance much closer to that of TrASPr. That is to be expected as well - Pangolin is constructed to capture changes in PSI (or splice site usage as defined by the authors), those changes are not even tissue specific for the CD19 data and the model has no problem/lack of capacity to generalize from the training set just like TrASPr does. In fact, if you only use combinations of known mutations seen during training a simple regression model gives correlation of ~92-95% (Cortés-López et al 2022). In summary, we believe that better understanding of what one can realistically expect from models such as SpliceAI, Pangolin, and TrASPr will go a long way to have them better understood and used effectively. We have tried to make this more clear in the revision.

(9) BOS seems like a separate contribution that belongs in a separate publication. Instead, consider providing more details on TrASPr.

We thank the reviewer for the suggestion. We agree those are two distinct contributions/algorithms and we indeed considered having them as two separate papers. However, there is strong coupling between the design algorithm (BOS) and the predictor that enables it (TrASPr). This coupling is both conceptual (TrASPr as a “teacher”) and practical in terms of evaluations. While we use experimental data (experiments done involving Daam1 exon 16, CD19 exon 2) we still rely heavily on evaluations by TrASPr itself. A completely independent evaluation would have required a high-throughput experimental system to assess designs, which is beyond the scope of the current paper. For those reasons we eventually decided to make it into what we hope is a more compelling combined story about generative models for prediction and design of RNA splicing.

(10) The authors should consider evaluating BOS using Pangolin or SpliceTransformer as the oracle, in order to measure the contribution to the sequence generation task provided by BOS vs TrASPr.

We can definitely see the logic behind trying BOS with different predictors. That said, as we note above most of BOS evaluations are based on the “teacher”. As such, it is unclear what value replacing the teacher would bring. We also note that given this limitation we focus mostly on evaluations in comparison to existing approaches (genetic algorithm or random mutations as a strawman).

**Recommendations for the authors:**

**Reviewer #1 (Recommendations for the authors):**
Additional comments:(1) Is your model picking up transcription factor binding sites in addition to RBPs? TFs have been recently shown to have a role in splicing regulation:

Daoud, Ahmed, and Asa Ben-Hur. "The role of chromatin state in intron retention: A case study in leveraging large scale deep learning models." PLOS Computational Biology 21.1 (2025): e1012755.

We agree this is an interesting point to explore, especially given the series of works from the Ben-Hur’s group. We note though that these works focus on intron retention (IR) which we haven’t focused on here, and we only cover short intronic regions flanking the exons. We leave this as a future direction as we believe the scope of this paper is already quite extensive.

(2) SpliceNouveau is a recently published algorithm for the splicing design problem:Wilkins, Oscar G., et al. "Creation of de novo cryptic splicing for ALS and FTD precision medicine." Science 386.6717 (2024): 61-69.

Thank you for pointing out Wilkins et al recent publication, we now refer to it as well.

(3) Please discuss the relationship between your model and this deep learning model. You will also need to change the following sentence: "Since the splicing sequence design task is novel, there are no prior implementations to reference."

We revised this statement and now refer to several recent publications that propose similar design tasks.

(4) I would suggest adding a histogram of PSI values - they appear to be mostly close to 1 or 0.

PSI values are indeed typically close to either 0 or 1. This is a known phenomenon illustrated in previous studies of splicing (e.g. Shen et al NAR 2012). We are not sure what is meant by the comment to add a histogram but we made sure to point this out in the main text:

“...Still, those statistics are dominated by extreme values, such that 33.2\% are smaller than 0.15 and 56.0\% are higher than 0.85. Furthermore, most cassette exons do not change between a given tissue pair (only 14.0\% of the samples in the dataset, \ie a cassette exon measured across two tissues, exhibit ΔΨ| ≥ 0.15).”

(5) Part of the improvement of TrASPr over Pangolin could be the result of a more extensive dataset.

Please see above responses and new analysis.

(6) In the discussion of the roles of alternative splicing, protein diversity is mentioned, but I suggest you also mention the importance of alternative splicing as a regulatory mechanism:Lewis, Benjamin P., Richard E. Green, and Steven E. Brenner. "Evidence for the widespread coupling of alternative splicing and nonsense-mediated mRNA decay in humans." Proceedings of the National Academy of Sciences 100.1 (2003): 189-192.

Thank you for the suggestion. We added that point and citation.

(7) Line 96: You use dPSI without defining it (although quite clear that it should be Delta PSI).

Fixed.

(8) Pretrained transformers: Have you trained separate transformers on acceptor and donor sites, or a single splice junction transformer?

Single splice junction pre-training.

(9) "TrASPr measures the probability that the splice site in the center of Se is included in some tissue" - that's not my understanding of what TrASPr is designed to do.

We revised the above sentence to make it more precise: “Given a genomic sequence context S_e_ = (s_e_,...,s_e_), made of a cassette exon *e* and flanking intronic/exonic regions, TrASPr predicts for tissue *c* the fraction of transcripts where exon *e* is included or skipped over, ΔΨ-_e,c,c’_.”

(10) Please include the version of the human genome annotations that you used.

We used GENCODE v40 human genome hg38- this is now included in the Data section.

(11) I did not see a description of the RBP-AE component in the methods section. A bit more detail on the model would be useful as well.

Please see above details about replacing RBP-AE with a simpler linear PCA “RBP-State” encoding. We added details about how the PCA was performed to the Methods section.

(12) Typos, grammar:- Fix the following sentence: ATP13A2, a lysosomal transmembrane cation transporter, linked to an early-onset form of Parkinson's Disease (PD) when 306 loss-of-function mutations disrupt its function.

Sentence was fixed to now read: “The first example is of a brain cerebellum-specific cassette exon skipping event predicted by TrASPr in the ATP13A2 gene (aka PARK9). ATP13A2 is a lysosomal transmembrane cation transporter, for which loss of function mutation has been linked to early-onset of Parkinson’s Disease (PD)”.

- Line 501: "was set to 4e−4"(the - is a superscript).

Fixed

- A couple of citations are missing in lines 580 and 581.

Thank you for catching this error. Citations in line 580, 581 were fixed.

(13) Paper title: Generative modeling for RNA splicing predictions and design - it would read better as "Generative modeling for RNA splicing prediction and design", as you are solving the problems of splicing prediction and splicing design.

Thank you for the suggestion. We updated the title and removed the plural form.

**Reviewer #2 (Recommendations for the authors):**
(1) Appendices are not very common in biology journals. It is also not clear what purpose the appendix serves exactly - it seems to repeat some of the things said earlier. Consider merging it into the methods or the main text.

We merged the appendices into the Methods section and removed redundancy.

(2) L112, "For instance, the model could be tasked with designing a new version of the cassette exon, restricted to no more than N edit locations and M total base changes." How are N and M different? Is there a difference between an edit location and a base change?

Yes, N is the number of locations (one can think of it as a start position) of various lengths (e.g. a SNP is of length 1) and the total number of positions edited is M. The text now reads “For instance, the model could be tasked with designing a new version of the cassette exon, restricted to no more than $N$ edit locations (\ie start position of one or more consecutive bases) and $M$ total base changes.”

(3) L122: "DEN was developed for a distinct problem". What prevents one from adapting DEN to your sequence design task? The method should be generic. I do not see what "differs substantially" means here. (Finally, wasn't DEN developed for the task you later refer to as "alternative splice site" (as opposed to "splice site selection")? Use consistent terminology. And in L236 you use "splice site variation" - is that also the same?).

Indeed, our original description was not clear/precise enough. DEN was designed and trained for two tasks: APA, and 5’ alternative splice site usage. The terms “selection”, “usage”, and “variation” were indeed used interchangeably in different locations and the reviewer was right, noting the lack of precision. We have now revised the text to make sure the term “relative usage” is used.

Nonetheless, we hold DEN was indeed defined for different tasks. See figures from Figure 2A, 6A of Linder et al 2020 (the reference was also incorrect as we cited the preprint and not the final paper):

In both cases DEN is trying to optimize a short region for selecting an alternative PA site (left) or a 5’ splice site (right). This work focused on an MPRA dataset of short synthetic sequences inserted in the designated region for train/test. We hold this is indeed a different type of data and task then the one we focus on here. Yes, one can potentially adopt DEN for our task, but this is beyond the scope of this paper. Finally, we note that a more closely related algorithm recently proposed is Ledidi (Schreiber et al 2025) which was posted as a pre-print. Similar to BOS, Ledidi tries to optimize a given sequence and adopt it with a few edits for a given task. Regardless, we updated the main text to make the differences between DEN and the task we defined here for BOS more clear, and we also added a reference to Ledidi and other recent works in the discussion section.

(4) L203, exons with DeltaPSI very close to 0.15 are going to be nearly impossible to classify (or even impossible, considering that the DeltaPSI measurements are not perfect). Consider removing such exons to make the task more feasible.

Yes, this is how it was done. As described in more details below, we defined changing samples as ones where the change was >= 0.15 and non-changing as ones where the change in PSI was < 0.05 to avoid ambiguous cases affecting the classification task.

(5) L230, RBP-AE is not explained in sufficient detail (and does not appear in the methods, apparently). It is not clear how exactly it is trained on each new cellular condition.

Please see response in the opening of this document and Q11 from

Reviewer 1

(6) L230, "significantly improving": the r value actually got worse; it is therefore not clear you can claim any significant improvement. Please mention that fact in the text.

This is a fair point. We note that we view the “a” statistic as potentially more interesting/relevant here as the Pearson “r” is dominated by points being generally close to 0/1. Regardless, revisiting this we realized one can also make a point that the term “significant” is imprecise/misplaced since there is no statistical test done here (side note: given the amount of points, a simple null of same distribution yes/no would pass significance but we don’t think this is an interesting/relevant test here). Also, we note that with the transition to PCA instead of RBP-AE we actually get improvements in both a and r values, both for the ENCODE samples shown in Figure 3a and the two new GTEX tissues we tested (see above). We now changed the text to simply state:

“...As shown in Figure 3a, this latent space representation allows TrSAPr to generalize from the six GTEX tissues to unseen conditions, including unseen GTEX tissues (top row), and ENCODE cell lines (bottom row). It improves prediction accuracy compared to TrASPr lacking PCA (eg a=88.5% vs a=82.3% for ENCODE cell lines), though naturally training on the additional GTEX and ENCODE conditions can lead to better performance (eg a=91.7%, for ENCODE, Figure 3a left column).”

(7) L233, "Notably, previous splicing codes focused solely on cassette exons", Rosenberg et al. focused solely on alternative splice site choice.

Right - we removed that sentence..

(8) L236, "trained TrASPr on datasets for 3' and 5' splice site variations". Please provide more details on this task. What is the input to TrASPr and what is the prediction target (splice site usage, PSI of alternative isoforms)? What datasets are used for this task?

The data for this data was the same GTEx tissue data processed, just for alternative 3’ and 5’ splice sites events. We revised the description of this task in the main task and added information in the Methods section. The data is also included in the repo.

(9) L243, "directly from genomic sequences", and conservation?

Yes, we changed the sentence to read “...directly from genomic sequences combined with related features”

(10) L262, what is the threshold for significant splicing changes?

The threshold is 0.15 We updated the main text to read the following:

The total number of mutations hitting each of the 1198 genomic positions across the 6106 sequences is shown in \FIG{mut_effect}b (left), while the distribution of effects ($|\Delta \Psi|$) observed across those 6106 samples is shown in \FIG{mut_effect}b (right). To this data we applied three testing schemes. The first is a standard 5-fold CV where 20\% of combinations of point mutations were hidden in every fold while the second test involved 'unseen mutation' (UM) where we hide any sample that includes mutations in specific positions for a total of 1480 test samples. As illustrated by the CDF in \FIG{mut_effect}b, most samples (each sample may involve multiple positions mutated) do not involve significant splicing changes. Thus, we also performed a third test using only the 883 samples were mutations cause significant changes ($|\Delta \Psi|\geq 0.15 $).

(11) L266, Pangolin performance is only provided for one of the settings (and it is not clear which). Please provide details of its performance in all settings.

The description was indeed not clear. Pangolin’s performance was similar to SpliceAI as mentioned above but retraining it on the CD19 data yielded much closer performance to TrASPr. We include all the matching tests for Pangolin after retraining in Figure 4 Supp Figure 1.

(12) Please specify "n=" in all relevant plots.

Fixed.

(13) Figure 3a, "The tissues were first represented as tokens, and new cell line results were predicted based on the average over conditions during training." Please explain this procedure in more detail. What are these tokens and how are they provided to the model? Are the cell line predictions the average of the predictions for the training tissues?

Yes, we compared to simply the average over the predictions for the training tissues for that specific event as baseline to assess improvements (see related work pointing for the need to have similar baselines in DL for genomics in https://pubmed.ncbi.nlm.nih.gov/33213499/). Regarding the tokens - we encode each tissue type as a possible value and feed the two tissues as two tokens to the transformer.

(14) Figure 4b, the total count in the histogram is much greater than 6106. Please explain the dataset you're using in more detail, and what exactly is shown here.

We updated the text to read:

“...we used 6106 sequence samples where each sample may have multiple positions mutated (\ie mutation combinations) in exon 2 of CD19 and its flanking introns and exons (Cortes et al 2022). The total number of mutations hitting each of the 1198 genomic positions across the 6106 sequences is shown in Figure 4b (left).”

(15) Figure 5a, how are the prediction thresholds (TrASPr passed, TrASPr stringent, and TrASPr very stringent) defined?

Passed: dpsi>0.1, Stringent: dpsi>0.15, Very stringent: dpsi>0.2 This is now included in the main text.

(16) L417, please include more detail on the relative size of TrASPr compared to other models (e.g. number of parameters, required compute, etc.).

SpliceAI is a general-purpose splicing predictor with 32-layer deep residual neural network to capture long-range dependencies in genomic sequences. Pangolin is a deep learning model specifically designed for predicting tissue-specific splicing with similar architecture as SpliceAI. The implementation of SpliceAI that can be found here https://huggingface.co/multimolecule/spliceai involves an ensemble of 5 such models for a total of ~3.5M parameters. TrASPr, has 4 BERT transformers (each 6 layers and 12 heads) and MLP a top of those for a total of ~189M parameters. Evo 2, a genomic ‘foundation’ model has 40B parameters, DNABERT has ~86M (a single BERT with 12 layers and 12 heads), and Borzoi has 186M parameters (as stated in https://www.biorxiv.org/content/10.1101/2025.05.26.656171v2). We note that the difference here is not just in model size but also the amount of data used to train the model. We edited the original L417 to reflect that.

(17) L546, please provide more detail on the VAE. What is the dimension of the latent representation?

We added more details in the Methods section like the missing dimension (256) and definitions for P(Z) and P(S).

(18) Consider citing (and possibly comparing BOS to) Ghari et al., NeurIPS 2024 ("GFlowNet Assisted Biological Sequence Editing").

Added.

(19) Appendix Figure 2, and corresponding main text: it is not clear what is shown here. What is dPSI+ and dPSI-? What pairs of tissues are you comparing? Spearman correlation is reported instead of Pearson, which is the primary metric used throughout the text.

The dPSI+ and dPSI- sets were indeed not well defined in the original submission. Moreover, we found our own code lacked consistency due to different tests executed at different times/by different people. We apologize for this lack of consistency and clarity which we worked to remedy in the revised version. To answer the reviewer’s question, given two tissues ($c,c'$), dPSI+ and dPSI- is for correctly classifying the exons that are significantly differentially included or excluded. Specifically, differential included exons are those for which $\Delta \Psi_{e,c1,c2} = \Psi_\Psi_{e,c1} - \Psi_{e,c2} \geq 0.15$, compared to those that are not ($\Delta \Psi_{e,c1,c2} < 0.05). Similarly, dPSI- is for correctly classifying the exons that are significantly differentially excluded in the first tissue or included in the second tissue ($\Delta \Psi_{e,c1,c2} = \Psi_\Psi_{e,c1} - \Psi_{e,c2} \leq -0.15$) compared to those that are not ($\Delta \Psi_{e,c1,c2} > -0.05). This means dPSI+ and dPSI- are dependent on the order of c1, c2. In addition, we also define a direction/order agnostic test for changing vs non changing events i.e. $|\Delta \Psi_{e,c1,c2}| \geq 0.15$ vs $|\Delta \Psi_{e,c1,c2}| < 0.05$. These test definitions are consistent with previous publications (e.g. Barash et al Nature 2010, Jha et al 2017) and also answer different biological questions: For example “Exons that go up in brain” and “Exons that go up in Liver” can reflect distinct mechanisms, while changing exons capture a model’s ability to identify regulated exons even if the direction of prediction may be wrong. The updated Appendix Figure 2 is now in the main text as Figure 2d and uses Pearson, while AUPRC and AUROC refer to the changing vs no-changing classification task described above such that we avoid dPSI+ and dPSI- when summarizing in this table over 3 pairs of tissues . Finally, we note that making sure all tests comply with the above definition also resulted in an update to Figure 2b/c labels and values, where TrASPr’s improvements over Pangolin reaches up to 1.8fold in AUPRC compared to 2.4fold in the earlier version. We again apologize for having a lack of clarity and consistent evaluations in the original submission.

(20) Minor typographical comments:- Some plots could use more polishing (e.g., thicker stroke, bigger font size, consistent style (compare 4a to the other plots)...).

Agreed. While not critical for the science itself we worked to improve figure polishing in the revision to make those more readable and pleasant.

- Consider using 2-dimensional histograms instead of the current kernel density plots, which tend to over-smooth the data and hide potentially important details.

We were not sure what the exact suggestion is here and opted to leave the plots as is.

- L53: dPSI_{e, c, c'} is never formally defined. Is it PSI_{e, c} - PSI_{e, c'} or vice versa?

Definition now included (see above).

- L91: Define/explain "transformer" and provide reference.

We added the explanation and related reference of the transformer in the introduction section and BERT in the method section.

- L94: exons are short. Are you referring here to the flanking introns? Please explain.

We apologize for the lack of clarity. We are referring to a cassette exon alternative splicing event as is commonly defined by the splice junctions involved that is from the 5’ SS of the upstream exon to the 3’ SS of the downstream exon. The text now reads:

“...In contrast, 24% of the cassette exons analyzed in this study span a region between the flanking exons' upstream 3' and downstream 5' splice sites that are larger than 10 kb.”

- L132: It's unclear whether a single, shared transformer or four different transformers (one for each splice site) are being pre-trained. One would at least expect 5' and 3' splice sites to have a different transformer. In Methods, L506, it seems that each transformer is pre-trained separately.We updated the text to read:“We then center a dedicated transformer around each of the splice sites of the cassette exon and its upstream and downstream (competing) exons (four separate transformers for four splice sites in total).”- L471: You explain here that it is unclear what tasks 'foundation' models are good for. Also in L128, you explain that you are not using a 'foundation' model. But then in L492, you describe the BERT model you're using as a foundation model!

Line 492 was simply a poor choice of wording as “foundation” is meant here simply as the “base component”. We changed it accordingly.

- L169, "pre-training ... BERT", explain what exactly this means. Is it using masking? Is it self-supervised learning? How many splice sites do you provide? Also explain more about the BERT architecture and provide references.

We added more details about the BERT architecture and training in the Methods section.

- L186 and later, the values for a and r provided here and in the below do not correspond to what is shown in Figure 2.

Fixed, thank you for noticing this.

- L187,188: What exactly do you mean by "events" and "samples"? Are they the same thing? If so, are they (exon, tissue) pairs? Please use consistent terminology. Moreover, when you say "changing between two conditions": do you take all six tissues whenever there is a 0.15 spread in PSI among them? Or do you take just the smallest PSI tissue and the largest PSI tissue when there is a 0.15 spread between them? Or something else altogether?Reviewer #2 is yet again correct that the definitions were not precise. A “sample” involves a specific exon skipping “event” measured in two tissues. The text now reads:“....most cassette exons do not change between a given tissue pair (only 14.0% of the samples in the dataset, i.e., a cassette exon measured across two tissues, exhibit |∆Ψ| ≥ 0.15). Thus, when we repeat this analysis only for samples involving exons that exhibited a change in inclusion (|∆Ψ| ≥ 0.15) between at least two tissues, performance degrades for all three models, but the differences between them become more striking (Figure 2a, right column).”- Figure 1a, explain the colors in the figure legend. The 3D effect is not needed and is confusing (ditto in panel C).

Color explanation is now added: “exons and introns are shown as blue rectangles and black lines. The blue dashed line indicates the inclusive pattern and the red junction indicates an alternative splicing pattern.”

These are not 3D effects but stacks to indicate multiple events/cases. We agree these are not needed in Fig1a to illustrate types of AS and removed those. However, in Fig1c and matching caption we use the stacks to indicate HT data captures many such LSVs over which ML algorithms can be trained.

- Figure 1b, this cartoon seems unnecessary and gives the wrong impression that this paper explores mechanistic aspects of splicing. The only relevant fact (RBPs serving as splicing factors) can be explained in the text (and is anyway not really shown in this figure).

We removed Figure 1b cartoon.

- Figure 1c, what is being shown by the exon label "8"?

This was meant to convey exon ID, now removed to simplify the figure.

- Figure 1e, left, write "Intron Len" in one line. What features are included under "..."? Based on the text, I did not expect more features.Also, the arrows emanating from the features do not make sense. Is "Embedding" a layer? I don't think so. Do not show it as a thin stripe. Finally, what are dPSI'+ and dPSI'-? are those separate outputs? are those logits of a classification task?

We agree this description was not good and have updated it in the revised version.

- Figure 1e, the right-hand side should go to a separate figure much later, when you introduce BOS.

We appreciate the suggestion. However, we feel that Figure 1e serves as a visual representation of the entire framework. Just like we opted to not turn this work into two separate papers (though we fully agree it is a valid option that would also increase our publication count), we also prefer to leave this unified visual representation as is.

- Figure 2, does the n=2456 refer to the number of (exons, tissues) pairs? So each exon contributes potentially six times to this plot? Typo "approximately".

The “n” refers to the number of samples which is a cassette event measured in two tissues. The same cassette event may appear in multiple samples if it was confidently quantified in more than two tissues. We updated the caption to reflect this and corrected the typo.

- Figure 2b, typo "differentially included (dPSI+) or excluded" .

Fixed.

- L221, "the DNABERT" => "DNABERT".

Fixed.

- L232, missing percent sign.

-

Fixed.

- L246, "see Appendix Section 2 for details" seems to instead refer to the third section of the appendix.

We do not have this as an Appendix, the reference has been updated.

- Figure 3, bottom panels, PSI should be "splice site usage"?

PSI is correct here - we hope the revised text/definitions make it more clear now.

- Figure 3b: typo: "when applied to alternative alternative 3'".

Fixed.

- p252, "polypyrimidine" (no capitalization).

Fixed.

- Strange capitalization of tissue names (e.g., "Brain-Cerebellum"). The tissue is called "cerebellum" without capitalization.

We used EBV (capital) for the abbreviation and lower case for the rest.

- Figure 4c: "predicted usage" on the left but "predicted PSI" on the right.

Right. We opted to leave it as is since Pangolin and SpliceAI do predict their definition of “usage” and not directly PSI, we just measure correlations to observed PSI as many works have done in the past.

- Figure 4 legend typo: "two three".

Fixed.

- L351, typo: "an (unsupervised)" (and no need to capitalize Transformer).

Fixed.

- L384, "compared to other tissues at least" => "compared to other tissues of at least".

Fixed.

- L549, P(Z) and P(S) are not defined in the text.

Fixed.

- L572, remove "Subsequently". Add missing citations at the end of the paragraph.

Fixed.

- L580-581, citations missing.

Fixed.

- L584-585, typo: "high confidince predictions"

Fixed.

- L659-660, BW-M and B-WM are both used. Typo?

Fixed.

- L895, "calculating the average of these two", not clear; please rewrite.

Fixed.

- L897, "Transformer" and "BERT", do these refer to the same thing? Be consistent.

BOS is a transformer and not a BERT but TrASPr uses the BERT architecture. BERT is a type of transformer as the reviewer is surely well aware so the sentence is correct. Still, to follow the reviewer’s recommendation for consistency/clarity we changed it here to state BERT.

- Appendix Figure 5: The term dPSI appears to be overloaded to also represent the difference between predicted PSI and measured PSI, which is inconsistent with previous definitions.

Indeed! We thank the reviewer again for their sharp eye and attention to details that we missed. We changed Supp Figure 5, now Figure 4 Supplementary Figure 2, to |PSI’-PSI| and defined those as the difference between TrASPr’s predictions (PSI’) and MAJIQ based PSI quantifications.